# ANCHORS AWEIGH! SAIL FOR OPTIMAL UNIFIED MULTI-MODAL REPRESENTATIONS

## ABSTRACT

A unified representation space in multi-modal learning is essential for effectively integrating diverse data sources, such as text, images, and audio, to enhance efficiency and performance across various downstream tasks. Recent binding methods, such as ImageBind (Girdhar et al., 2023), typically rely on a single, fixed anchor modality for aligning multi-modal data. We mathematically analyze these **fixed anchor binding methods** and uncover significant limitations: (1) over-reliance on the choice of the anchor modality, (2) inadequate capture of intra-modal information, and (3) failure to account for cross-modal correlation among non-anchored modalities. To address these issues, we propose the need for **adaptive anchor binding methods**, exemplified by our framework CENTROBIND. The proposed method uses adaptively adjustable centroid-based anchors generated from all available modalities, leading to a balanced and rich representation space. We theoretically demonstrate that our approach captures three critical properties of multi-modal learning—intra-modal learning, inter-modal learning, and multi-modal alignment—while constructing a unified representation that spans all modalities. Experiments on both synthetic and real-world datasets show that adaptive anchor methods such as CENTROBIND consistently outperform fixed anchor binding methods, verifying our analysis.

## 1 INTRODUCTION

Multi-modal alignment is defined as identifying and exploiting relationships and correspondences between multiple modalities (e.g., text, image, audio) viewing common phenomena to establish meaningful connections between their representations (Baltrušaitis et al., 2018). A common approach is learning a shared embedding space (Tu et al., 2022; Girdhar et al., 2023; Liang et al., 2024b; Zhu et al., 2024), which aims to project data from multiple modalities into a common embedding space by clustering similar items together for direct comparison and linkage. This approach leverages well-trained single-modal embeddings, aligning them with auxiliary objective functions like contrastive loss (Oord et al., 2018) or triplet loss (Schroff et al., 2015) to minimize distances between similar items and maximize distances between dissimilar ones across modalities.

Instead of training separate models for each modality, ImageBind (Girdhar et al., 2023) pairs **images** with other modalities and projects them into a common image embedding space. Similarly, (Zhu et al., 2024) shows that pairing **texts** with other modalities (LanguageBind) improves cross-modal retrieval performance when language serves as the anchor modality. This approach has inspired various "-Bind" methods tailored to align different modalities for specific domains, such as molecular modeling (Xiao et al., 2024), medical imaging (Gao et al., 2024), brain signals (Yang et al., 2024b), and music selection for videos (Teng et al., 2024). These models commonly use image or text as the **anchor embedding** due to the abundance of data, with other modalities projected into this anchor representation.

We refer to these approaches as **Fixed Anchor Bind** (FABIND) methods, where the primary anchor modality's embedding space is held fixed while aligning others. Many "-Bind" methods maximize mutual information $I(\mathbf{Z}_1; \mathbf{Z}_i)$ between the anchor representation $\mathbf{Z}_1$ and each non-anchor $\mathbf{Z}_i$ for $i \in 2, \ldots, M$. Despite their practical appeal for unified multimodal representations, we show, both theoretically and empirically, that they have important limitations.

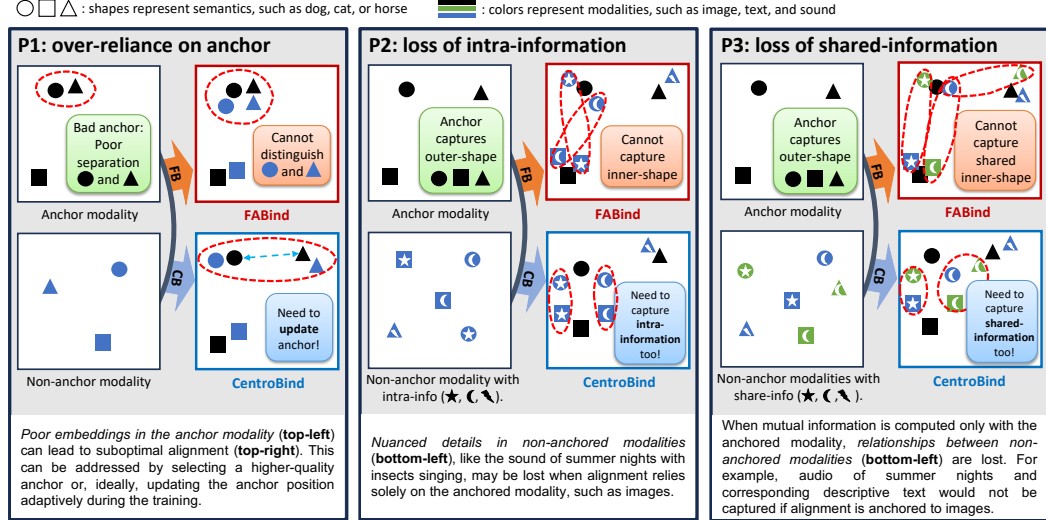

Figure 1: Limitations of fixed-anchor binding (FABIND) and remedy via CentroBind (CB). Shapes denote semantic classes (e.g., dog, cat, horse) and colors denote modalities (image, text, audio). **P1–Over-reliance on anchor:** a poor or poorly chosen anchor yields misalignment across modalities. **P2–Loss of intra-modal information:** anchoring suppresses modality-specific cues present only in non-anchored views. **P3–Loss of shared information among non-anchors:** optimizing only anchor ↔ others ignores correlations between non-anchored modalities. CB addresses all three by computing adaptive, batch-wise anchors (centroids) from the available modalities and aligning each modality to this shared anchor, preserving intra-information and non-anchor shared-information while improving overall alignment.

**Issues with fixed anchor binding.** As illustrated in Figure 1, we discover three limitation of FABIND. **P1–over-reliance on anchor:** the best anchor choice depends on embedding quality and task, and common defaults (image or text) can be suboptimal when no single modality dominates. **P2–loss of intra-information:** a fixed anchor can discard semantics captured primarily by other modalities (e.g., sound revealing mood, images conveying expression beyond text like "a dog barks loudly"). **P3–loss of shared-information:** optimizing only anchor–other pairs ignores complementarities among non-anchor modalities. These issues motivate adaptive anchor alignment as an alternative to FABIND; we formalize FABIND's deficiencies in Section 2.

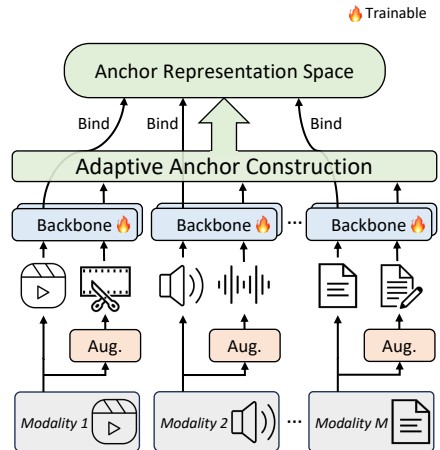

Figure 2: Graphical illustration of adaptive anchor alignment: Adaptive anchors are dynamically generated from current embeddings using an aggregation method for every batch.

**Adaptive anchor alignment.** We propose an alternative to fixed anchor alignment by replacing fixed anchors with *"adaptive"* anchors computed from paired multi-modal samples. Our proposed method, CEN-TROBIND, an example of adaptive anchor bind methods, described in Section 3, removes the need for selecting a fixed anchor modality, instead calculates the centroid over the aggregate of all modality's representations and generates a multi-modal anchor representation, as shown in Figure 2. We note that various aggregation methods can be used for these embeddings, such as computing a weighted average when the relative importance or quality of the backbone models is known, or even learning the weight coefficients dynamically during training. Some of these options are explored, with experimental results provided in Appendix C.1.

Our theoretical analysis demonstrates that CENTROBIND effectively addresses three critical components of multi-modal learning: 1) capturing intra-modal mutual information, 2) learning inter-modal mutual information, and 3) performing multi-modal alignment by maximizing embedding similarity measures. By incorporating these elements, CENTROBIND outperforms other multi-modal alignment methods, as shown empirically on both synthetic and real-world datasets in various downstream tasks. The proposed approach yields a unified representation space and, in the perspective of (Huh et al., 2024), who contends that multi-modal representations better align as they move toward a platonic representation that captures the semantic information of all modalities simultaneously.

## 2 PROBLEM FORMULATION

In this section, we describe general representation learning and binding problems in multi-modal learning. Then, we analyze fixed-anchor-bind (FABIND) methods such as ImageBind (Girdhar et al., 2023), that bind multi-modal representations to a user-selected fixed modality.

### 2.1 REPRESENTATION LEARNING FRAMEWORK

**Notation.** Boldface upper case letters (e.g., $\mathbf{X}$) denote random vectors, and a realization is denoted by the boldface lower case letters (e.g., $\boldsymbol{x}$); For $n \in \mathbb{N}$, $[n] := \{1, 2, \cdots, n\}$; $P_{\mathbf{X}}$ and $P_{\mathbf{X},\mathbf{Y}}$ denote the marginal and the joint distributions of $\mathbf{X}$ and $(\mathbf{X}, \mathbf{Y})$, respectively.

Given $M$ datasets $\mathcal{D} = \{\mathcal{D}_i\}_{i=1}^{M}$, let $\mathcal{D}_i = \{(\boldsymbol{x}_{i,j}, \boldsymbol{y}_{i,j})\}_{j=1}^{N_i}$ be the dataset from the $i$-th modality, where $\boldsymbol{x}_{i,j} \in \mathcal{X}_i$ and $\boldsymbol{y}_{i,j} \in \mathcal{Y}_i$ are respectively the $j$-th input instance (e.g., feature vector) and the corresponding label in $i$-th modality, and we assume that $(\boldsymbol{x}_{i,j}, \boldsymbol{y}_{i,j}) \overset{\text{i.i.d.}}{\sim} P_{\mathbf{X}_i,\mathbf{Y}_i}$.[1] We assume that $j$ indexes paired samples among modalities. For instance, $\boldsymbol{x}_{1,c}$ and $\boldsymbol{x}_{2,c}$ are features having similar semantic information (e.g., dog image and dog sound) in $\mathcal{D}_1$ and $\mathcal{D}_2$. The goal of representation learning is to build $M$ encoders $f_i : \mathcal{X}_i \to \mathcal{Z}_i$ for each modality, which maps the input instances $\boldsymbol{x}_{i,j}$ to its embedding $\boldsymbol{z}_{i,j} = f_i(\boldsymbol{x}_{i,j})$, preserving as much information about $\boldsymbol{x}_{i,j}$ as possible.

For the uni-modal case ($M = 1$), keeping maximum information about $\boldsymbol{x}_{1,j}$ at its embedding $\boldsymbol{z}_{1,j}$ is generally preferred based on the "InfoMax" principle (Linsker, 1988), under which the objective is to maximize mutual information $I(\mathbf{X}_i; f(\mathbf{X}_i))$ between $\mathbf{X}_i$ and $f(\mathbf{X}_i)$. Throughout the paper, we call $I(\mathbf{X}_i; f(\mathbf{X}_i))$ *intra information* on $\mathbf{X}_i$. For the multi-modal case ($M \geq 2$), on top of the InfoMax principle, "minimal sufficiency" is proposed in (Tian et al., 2020b), which suggests maximizing *shared information* $I(f_i(\mathbf{X}_i); f_l(\mathbf{X}_l))$ between $f_i(\mathbf{X}_i)$ and $f_l(\mathbf{X}_l)$, while minimizing the *unique information* $I(\mathbf{X}_i; f_i(\mathbf{X}_i)|\{\mathbf{X}_l\}_{l \neq i})$. Although minimal sufficiency often leads to an efficient encoder with better performance in numerous multi-modal downstream tasks, it is not always a good strategy as there exist exceptions where the unique information on an individual modality is crucial (Liang et al., 2024b; Wang et al., 2022a). In other words, the optimality of minimal sufficiency is task-dependent. To avoid task dependency, we do not consider minimal sufficiency; instead, we maximize intra and shared information without reducing unique information. Next, we formalize the notion of sufficient embedding.

**Definition 2.1** ($\mathcal{Z}_i$-Sufficient embedding of $\mathbf{X}_i$ for $\mathbf{X}_l$). For an embedding space $\mathcal{Z}_i$, the embedding $f_i(\mathbf{X}_i)$ is $\mathcal{Z}_i$-sufficient for $\mathbf{X}_l$ if and only if the embedding $f_i(\mathbf{X}_i)$ achieves the maximum mutual information between $f_i(\mathbf{X}_i)$ and $\mathbf{X}_i$. Specifically,

$$f_i \in \arg \max_{f: \mathcal{X}_i \to \mathcal{Z}_i} I(f(\mathbf{X}_i); \mathbf{X}_l). \tag{1}$$

We call $f_i$ sufficient encoder of $\mathbf{X}_i$ for $\mathbf{X}_l$.

We note that if $i = l$, the sufficient encoder provides embeddings with maximum intra information, and if $i \neq l$, it gives embeddings with maximum shared information between $i$-th and $l$-th modalities.[2]

In the context of contrastive representation learning having the goal of attaining sufficient encoders in Definition 2.1, InfoNCE loss $I_{\text{NCE}}(X; Y)$ is often employed since it relates to mutual information. Specifically, InfoNCE provides a lower bound on mutual information, i.e.,

---

[1]In self-supervised learning, labels might not exist, which corresponds to the case that $\boldsymbol{y}_{i,j}$ are empty.

[2]With $\mathcal{Z}_i$ such that $\max_{f: \mathcal{X}_i \to \mathcal{Z}_i} I(f(\mathbf{X}_i); \mathbf{X}_l) = I(\mathbf{X}_i; \mathbf{X}_l)$, Definition 2.1 says that $f_i(\boldsymbol{x}_{i,j})$ is a sufficient statistic (Polyanskiy & Wu, 2024) of $\boldsymbol{x}_{i,j}$ for $\boldsymbol{x}_{l,j}$ as the encoding entails no information loss.

$I(\mathbf{X}; \mathbf{Y}) \geq -I_{\mathrm{NCE}}(\mathbf{X}; \mathbf{Y})$ (Oord et al., 2018), and thus minimizing InfoNCE leads to an increase in mutual information. InfoNCE loss between embeddings $\mathbf{U}$ and $\mathbf{V}$ can be written as follows:

$$I_{\mathrm{NCE}}(\mathbf{U}; \mathbf{V}|\tau) = \mathbb{E}_P \left[ -\log \frac{\exp(\mathbf{U}^\top \mathbf{V}/\tau)}{\exp(\mathbf{U}^\top \mathbf{V}/\tau) + \sum_{i=1}^N \exp(\mathbf{U}^\top \mathbf{V}_i/\tau)} \right], \tag{2}$$

where the expectation is taken with respect to the distribution $P = P_{\mathbf{U},\mathbf{V}} \prod_{i=1}^N P_{\mathbf{V}_i}$. Here, we say $(\mathbf{U}, \mathbf{V})$ is a positive pair if $(\mathbf{U}, \mathbf{V}) \sim P_{\mathbf{U},\mathbf{V}}$ and $(\mathbf{U}, \mathbf{V})$ is a negative pair if $(\mathbf{U}, \mathbf{V}_i) \sim P_{\mathbf{U}} P_{\mathbf{V}_i}$. In (2), $N \geq 1$ and $\tau > 0$ are hyper-parameters, specifying the number of negative samples and the temperature parameter. For simplicity, in this paper, we assume that embeddings are normalized (Wang & Isola, 2020) to unit vectors and are of the same dimensionality. Then, the exponent $\mathbf{U}^\top \mathbf{V}/\tau$ in (2) is proportional the cosine similarity score between $\mathbf{U}$ and $\mathbf{V}$.

## 2.2 BINDING REPRESENTATION SPACES

In addition to the objective of capturing intra and shared information, multi-modal learning often takes into account multi-modal alignment (Radford et al., 2021; Duan et al., 2022). Without multi-modal alignment, each modality can only access its own embedding structure depending on its encoder. For example, embeddings of cat and dog images, respectively, locate around $(1, 0)$ and $(0, 2)$ in $\mathbb{R}^2$, whereas embeddings of cat and dog text can lie around $(0, 2)$ and $(1, 0)$. Such a misalignment can happen even for sufficient encoders (Definition 2.1), since the mutual information is invariant to one-to-one mappings (Polyanskiy & Wu, 2024).

To align multi-modal embedding spaces, a unified representation space (Radford et al., 2021; Zhou et al., 2023) or multi-modal alignment (Wang et al., 2023; Liang et al., 2024c) have been proposed for multi-modal representation learning, in which embeddings of multi-modal features having similar semantic should near each other in embedding space. Several FABIND methods have been proposed (see Appendix A for a summary of FABIND methods) that include ImageBind (Girdhar et al., 2023). ImageBind sets the image modality as the fixed anchor modality, and then InfoNCE loss is minimized between the embeddings of the anchor modality and the other modalities. FABIND (e.g., ImageBind) aims to find encoders $f_i^{\mathrm{FB}}$ for all modalities, except the anchor modality, such that for all $i \in \{2, \cdots, M\}$,

$$f_i^{\mathrm{FB}} \in \arg \min_{f_i: \mathcal{X}_i \to \mathcal{Z}_i} I_{\mathrm{NCE}}(f_1(\mathbf{X}_1); f_i(\mathbf{X}_i)), \tag{3}$$

where $f_1$ is the encoder for the anchor modality (an image encoder in ImageBind). Note that FABIND freezes $f_1$, initialized by an existing pretrained model, during the optimization.

## 2.3 ANALYSIS OF FABIND

In this section, we characterize the theoretical limitations of FABIND. To this end, we rewrite (3) as

$$f_i^{\mathrm{FB}} \in \arg \max_{f_i: \mathcal{X}_i \to \mathcal{Z}_i} I(f_1(\mathbf{X}_1); f_i(\mathbf{X}_i)), \ \forall i \in \{2, \cdots, M\}, \tag{4}$$

reflecting the fact that minimizing InfoNCE loss is equivalent to maximizing mutual information.[3] Let FABIND encoders from (4) for each modality be defined as $\mathcal{F}^{\mathrm{FB}} = \{f_1, f_2^{\mathrm{FB}}, \cdots, f_M^{\mathrm{FB}}\}$. The anchor encoder $f_1$ is fixed during the entire FABIND procedure. Moreover, we assume that $I(f_1(\mathbf{X}_1); f_i^{\mathrm{FB}}(\mathbf{X}_i)) = I(f_1(\mathbf{X}_1); \mathbf{X}_i)$ is the maximum value that can be achieved by (4) due to data processing inequality (Polyanskiy & Wu, 2024). We next demonstrate that the quality of anchor embedding $f_1(\mathbf{X}_1)$ significantly impacts the performance of $\mathcal{F}^{\mathrm{FB}}$ in terms of shared information. The following propositions show the dependency of FABIND on anchor embedding quality.

**Proposition 2.2** (FABIND with sufficient anchor). *Let $f_1^{\mathrm{suf}}(\mathbf{X}_1)$ be a sufficient embedding of the anchor $\mathbf{X}_1$, and let $\mathbf{X}_i, i \in [M]$, be a discrete random variable. Assume that $f_i^{\mathrm{FB}}, i \in \{2, \cdots, M\}$ are obtained by (4) with a sufficient anchor encoder $f_1 = f_1^{\mathrm{suf}}$, i.e., $I(f_1^{\mathrm{suf}}(\mathbf{X}_1); f_i^{\mathrm{FB}}(\mathbf{X}_i)) = I(f_1^{\mathrm{suf}}(\mathbf{X}_1); \mathbf{X}_i)$. Then, for all $i \in \{2, \cdots, M\}$,*

$$I(f_1^{\mathrm{suf}}(\mathbf{X}_1); f_i^{\mathrm{FB}}(\mathbf{X}_i)) = I(\mathbf{X}_1; \mathbf{X}_i). \tag{5}$$

---

[3]In contrast to (3), $f_i^{\mathrm{FB}}$ in (4) might not be aligned with other modalities due to the one-to-one mapping invariant property of mutual information. However, we do not analyze the multi-modal alignment of FABIND from (4), but rather investigate the performance of encoders in terms of the sufficiency in Definition 2.1.

*Proof.* The proof is in Appendix B.1. □

**Proposition 2.3** (FABIND with insufficient anchor). *Let $f_1^{\text{ins}}(\mathbf{X}_1)$ be an insufficient embedding of the anchor $\mathbf{X}_1$ for $\mathbf{X}_1$, in the sense that there exists some $\epsilon > 0$ such that $I(f_1^{\text{ins}}(\mathbf{X}_1); \mathbf{X}_1) < \epsilon \leq \max_f I(f(\mathbf{X}_1); \mathbf{X}_1)$. Assume that $f_i^{\text{FB}}, i \in \{2, \cdots, M\}$ are obtained by (4) with $f_1 = f_1^{\text{ins}}$, i.e., $I(f_1^{\text{ins}}(\mathbf{X}_1); f_i^{\text{FB}}(\mathbf{X}_i)) = I(f_1^{\text{ins}}(\mathbf{X}_1); \mathbf{X}_i)$. Then,*

$$I(f_1^{\text{ins}}(\mathbf{X}_1); f_i^{\text{FB}}(\mathbf{X}_i)) < \epsilon, \ \forall i \in \{2, \cdots, M\}. \tag{6}$$

*Proof.* The proof is in Appendix B.2. □

Proposition 2.2 shows that the FABIND encoders $\mathcal{F}^{\text{FB}}$ learned with a sufficient anchor embedding can achieve the maximum shared information between the anchor and the other modalities. However, it does not guarantee shared information between *non-anchored* modalities $I(f_i(\mathbf{X}_i); f_l(\mathbf{X}_l)), \ i, l \neq 1$, which can also be seen from (4). Proposition 2.3 establishes that an insufficient anchor may lead to a reduction of shared information between the anchor and the other modalities, implying that the performance of FABIND may overly depend on the quality of the arbitrarily selected anchor. Next, we enumerate three limitations in FABIND, as illustrated in Figure 1:

**P1–Over-reliance on a single anchor modality.** Achieving maximum shared information requires sufficient anchor representation (Proposition 2.2 and 2.3), which depends on having both an informative modality and a sufficient encoder. Without these conditions, FABIND may not effectively capture shared information.

**P2–Failure to capture intra information.** Even with sufficient anchor representation, FABIND may not provide encoders with maximum intra information. This is because its objective function (4) does not take into account the mutual information between a sample from a modality and its augmentation, i.e., $I(f_i(\mathbf{X}_i); \mathbf{X}_i)$.

**P3–Absence of shared information among non-anchored modalities.** The objective function of FABIND (4) focuses solely on learning shared information between anchor and non-anchored modalities, while disregarding shared information *among non-anchored modalities*. This implies that FABIND may not capture shared information among them. This limitation could render FABIND less effective when crucial shared information exists among non-anchored modalities.

We identify three key limitations (**P1**, **P2**, and **P3**) of FABIND, supported by empirical evidence presented in Section 4.1. In addition to these constraints, the representations generated by FABIND may not approximate an ideal standard, such as the "Platonic representation" described in (Huh et al., 2024). Achieving an optimal multi-modal representation necessitates the comprehensive integration of all modalities to fully capture their information. However, FABIND falls short of this objective. To address these shortcomings, we next introduce an alternative multi-modal representation that fully leverages sample-level information across all modalities.

## 3 ALIGNMENT USING ADAPTIVE ANCHOR

To address the limitations of fixed modality anchoring, we propose the concept of **adaptive anchor alignment**. We aim for a unified representation that (1) remains unbiased toward any single modality and (2) achieves high similarity alignment across *all* modalities. In this paper, we introduce a centroid-based anchor representation that naturally fulfills these two requirements by acting as a geometric center. Specifically, we train the encoders by minimizing the InfoNCE loss between each modality and the centroid (similar to FABIND). Although other adaptive anchor methods—such as median or weighted average—are viable alternatives (see Appendix C.1), we focus on the centroid for its simplicity and strong theoretical grounding in aligning multi-modal representations. Next, we formally define CENTROBIND and show how it maximizes both intra-modal and cross-modal information within a unified embedding space.

### 3.1 CENTROBIND

Consider $M$ modalities with corresponding encoders $\{f_i\}_{i=1}^M$. The CENTROBIND algorithm is presented in Algorithm 1 in Appendix C, and a graphical illustration is given in Figure **??**. In the following, we describe each step of CENTROBIND.

**Initial encoders.** We initialize $M$ encoders $f_i : \mathcal{X}_i \to \mathcal{Z}$, $\forall i \in [M]$ for the $M$ modalities. These encoders can either be pretrained models (i.e., backbones) or parameterized models with random weights. The primary constraint for these initial encoders is that their output space must be the modality-independent embedding space $\mathcal{Z}$. When using pretrained encoders that produce embeddings in different output spaces, these are projected onto the embedding space $\mathcal{Z}$, ensuring consistency of output space across modalities.

**Anchor embedding.** Recall that $\boldsymbol{x}_{i,j} \in \mathcal{D}_i$ denotes the $j$-th feature in the $i$-th modality, where $j$ indexes positive pairs of features (e.g., different views of the same object). In each training iteration of CENTROBIND, we need to compute an anchor embedding $\boldsymbol{a}_j$ for the $j$-th multi-modal positive features $\{\boldsymbol{x}_{i,j}\}_{i=1}^M$. This anchor $\boldsymbol{a}_j$ serves as a desirable aligned embedding for these features. The anchor $\boldsymbol{a}_j$ is calculated as follows:

$$\boldsymbol{a}_j = \text{mean}\left(\{f_i(\boldsymbol{x}'_{i,j})\}_i\right), \tag{7}$$

where $\text{mean}(\cdot)$ denotes the mean operator that computes the average of its input, and $\boldsymbol{x}'_{i,j}$ represents an augmented version of $\boldsymbol{x}_{i,j}$. If $\{\boldsymbol{x}_{i,j}\}_{i=1}^M$ are available in multi-modal datasets, the anchor is given by $\boldsymbol{a}_j = \frac{1}{M}\sum_{i=1}^M f_i(\boldsymbol{x}'_{i,j})$. If only $m < M$ positive pairs are present among $M$ modalities, the anchor is given by $\boldsymbol{a}_j = \frac{1}{m}\sum_{i\in\mathcal{I}_j} f_i(\boldsymbol{x}'_{i,j})$, where $\mathcal{I}_j$ is the set of indices of modalities having the $m$ available features.

**Binding encoders to the anchor.** Once anchor embeddings $\{\boldsymbol{a}_j\}_j$ are derived from a batch $B = \{\boldsymbol{x}_{i,j}\}_{i,j}$, CENTROBIND aligns each modality-specific encoder embedding with the anchor embedding by minimizing InfoNCE loss. Specifically, let $\mathbf{A} = \text{mean}(\{f_i(\mathbf{X}_i)\}_i)$ represent the anchor embedding variable. Then, CENTROBIND aims to minimize InfoNCE loss $I_{\text{NCE}}(\mathbf{A}; f_i(\mathbf{X}_i))$ across all modalities $i \in [M]$. A detailed expression for this loss is provided in (8).

CENTROBIND optimizes the following symmetrized loss function:

$$\mathcal{L}_{\text{CB}}(f_i|\tau) = I_{\text{NCE}}(\mathbf{A}; f_i(\mathbf{X}_i)|\tau) + I_{\text{NCE}}(f_i(\mathbf{X}_i); \mathbf{A}|\tau), \tag{8}$$

where $\mathcal{L}_{\text{CB}}(f_i|\tau)$ denotes the loss function for the $i$-th modality. In particular, with a batch data $B = \{\boldsymbol{x}_{i,j} : i \in [M], j \in \mathcal{I}_B\}$, the loss can be computed as

$$I_{\text{NCE}}(\mathbf{A}; f_i(\mathbf{X}_i)|\tau) = \frac{-1}{|\mathcal{I}_B|}\sum_{k=1}^{|\mathcal{I}_B|} \log \frac{\exp(\boldsymbol{a}_k^\top f_i(\boldsymbol{x}_{i,k})/\tau)}{\sum_{j\in\mathcal{I}_B}\exp(\boldsymbol{a}_k^\top f_i(\boldsymbol{x}_{i,j})/\tau)} \tag{9a}$$

$$I_{\text{NCE}}(f_i(\mathbf{X}_i); \mathbf{A}|\tau) = \frac{-1}{|\mathcal{I}_B|}\sum_{k=1}^{|\mathcal{I}_B|} \log \frac{\exp(\boldsymbol{a}_k^\top f_i(\boldsymbol{x}_{i,k})/\tau)}{\sum_{j\in\mathcal{I}_B}\exp(f_i^\top(\boldsymbol{x}_{i,k})\boldsymbol{a}_j/\tau)}. \tag{9b}$$

### 3.2 THEORETICAL ANALYSIS OF CENTROBIND

We start by providing a lower bound on the objective function of CENTROBIND $\mathcal{L}_{\text{CB}}(f_i|\tau)$ (8) in Theorem 3.1, followed by an analysis of the minimizer of $\mathcal{L}_{\text{CB}}(f_i|\tau)$.

**Theorem 3.1.** *Consider $B = \{\boldsymbol{x}_{i,j} : i \in [M], j \in \mathcal{I}_B\}$ with a set of indices $\mathcal{I}_B$, where $\boldsymbol{x}_{i,j}$ is the $j$-th sample of $i$-th modality. Then, for any encoders $\{f_i\}_i$ and for any $\tau > 0$, (9a) is bounded as*

$$|\mathcal{I}_B| I_{\text{NCE}}\left(\mathbf{A}; f_i(\mathbf{X}_i)\mid\tau\right) \geq \sum_{l=1}^M I_{\text{NCE}}\left(f_l(\mathbf{X}'_l); f_i(\mathbf{X}_i)\,\bigg|\,\frac{\tau M}{|\mathcal{I}_B|}\right) - \sum_{k=1}^{|\mathcal{I}_B|}\log C_{\mathcal{F},k,i}, \tag{10}$$

*where $C_{\mathcal{F},k,i} = \frac{(c_{\mathcal{F},k,i}^{\min} + c_{\mathcal{F},k,i}^{\max})^2}{4c_{\mathcal{F},k,i}^{\min} c_{\mathcal{F},k,i}^{\max}}$ with $g(l,j|k,i) := \exp\left(\frac{|\mathcal{I}_B|f_l^\top(\boldsymbol{x}'_{l,k})f_i(\boldsymbol{x}_{i,j})}{\tau M}\right)$,*

$$c_{\mathcal{F},k,i}^{\min} = \min_{l\in[M],j\in\mathcal{I}_B} g(l,j|k,i), \quad \text{and} \quad c_{\mathcal{F},k,i}^{\max} = \max_{l\in[M],j\in\mathcal{I}_B} g(l,j|k,i). \tag{11}$$

*Proof.* The proof is in Appendix B.3. □

Theorem 3.1 provides a lower bound of $I_{\text{NCE}}(\mathbf{A}; f_i(\mathbf{X}_i) \mid \tau)$ in (9a), which is a part of the CEN-TROBIND objective function $\mathcal{L}_{\text{CB}}(f_i|\tau)$. Thus CENTROBIND minimizes a lower bound (10) that consists of two terms, $\sum_{l=1}^{M} I_{\text{NCE}}\left(f_l(\mathbf{X}_l'); f_i(\mathbf{X}_i) \mid \frac{\tau M}{|\mathcal{I}_B|}\right)$ and $-\sum_{k=1}^{|\mathcal{I}_B|} \log C_{\mathcal{F},k,i}$. We next provide intuition on why a minimization of the lower bound is justified.

**The effect of minimizing $\sum_{l=1}^{M} I_{\text{NCE}}(f_l(\mathbf{X}_l'); f_i(\mathbf{X}_i) \mid \cdot)$.** The objective of minimizing $\sum_{l=1}^{M} I_{\text{NCE}}(f_l(\mathbf{X}_l'); f_i(\mathbf{X}_i) \mid \frac{\tau M}{|\mathcal{I}_B|})$ is to reduce several InfoNCE losses. Here, each term in the sum represents InfoNCE loss between embeddings $f_l(\mathbf{X}_l')$ from modality $l$ and $f_i(\mathbf{X}_i)$ from modality $i$, with $\frac{\tau M}{|\mathcal{I}_B|}$ being a temperature parameter for scaling the loss. This summation can be divided into two components: 1) Intra Information: When $l = i$, the term measures the similarity between embeddings within the same modality. Minimizing this loss enhances the representation of modality $i$, improving intra information; 2) Shared Information: When $l \neq i$, the term measures the similarity between embeddings from different modalities. Minimizing these losses helps in learning shared information between modalities, contributing to a more representative multi-modal embedding.

By optimizing this summation, CENTROBIND effectively captures both intra and shared information. As shown below, this generally results in a more balanced representation for the modalities. In contrast, as noted in Section 2.3, FABIND does not adequately capture intra information and shared information between non-anchored modalities. This limitation highlights the advantage of CENTROBIND in achieving a more integrated multi-modal representation than fixed anchor binding methods.

**The effect of minimizing $-\sum_{k=1}^{|\mathcal{I}_B|} \log C_{\mathcal{F},k,i}$.** We show the effect of growing $C_{\mathcal{F},k,i}$ in terms of cosine similarity score between embeddings. Since $C_{\mathcal{F},k,i} = \frac{1}{4}\left(\sqrt{\gamma} + \sqrt{\frac{1}{\gamma}}\right)^2$ with $\gamma = \frac{c_{\mathcal{F},k,i}^{\max}}{c_{\mathcal{F},k,i}^{\min}} \geq 1$, maximizing $C_{\mathcal{F},k,i}$ is equivalent to simultaneously maximizing $c_{\mathcal{F},k,i}^{\max}$ and minimizing $c_{\mathcal{F},k,i}^{\min}$. For ease of analysis, we assume that the encoders are reasonably well-trained. Then, since a positive pair of embeddings normally yields higher similarity score, $c_{\mathcal{F},k,i}^{\max}$ is attained by choosing $l = i$ and $j = k$ in (11) as such choices make $\boldsymbol{x}_{l,k}'$ be positive pair with $\boldsymbol{x}_{i,j}$. Thus, $c_{\mathcal{F},k,i}^{\max}$ is roughly proportional to the similarity score of a positive pair of embeddings. Conversely, $c_{\mathcal{F},k,i}^{\min}$ corresponds to the similarity scores of negative pairs, which tend to be low. Hence, minimizing $-\sum_{k=1}^{|\mathcal{I}_B|} \log C_{\mathcal{F},k,i}$ enhances the similarity scores for positive pairs and reduces those for negative pairs, improving the overall multi-modal alignment.

These comments suggest that CENTROBIND addresses the limitations **P1**, **P2**, and **P3** of FABIND identified in Section 2.3.

## 4 EXPERIMENT

To thoroughly evaluate the effectiveness of the proposed method, we design two sets of experiments: **(1)** experiments on *synthetically* generated datasets and **(2)** experiments on *real-world* datasets spanning diverse modality domains. The synthetic datasets allow **controlled testing of extreme scenarios, such as varying numbers of modalities, modality imbalance, and backbone quality.** The real-world experiments demonstrate that the proposed method generalizes well **across datasets of different scales:** DreamBooth (Ruiz et al., 2023) (∼180 images across 30 subjects), MUStARD (Castro et al., 2019) (∼690 labeled clips), AVE (Tian et al., 2018) (∼4,143 videos across 28 event categories), UR-FUNNY (Hasan et al., 2019) (∼1,866 TED videos from 1,741 speakers, totaling ∼90 hours), and AudioSet (Gemmeke et al., 2017) (∼2 million human-labeled 10-second clips across 632 event classes). We compare CENTROBIND, FABIND  (anchored at the text modality for MUStARD and at the image modality for the other datasets), UniBind (Lyu et al., 2024), AudioCLIP (Guzhov et al., 2022), and ViT-Lens (Lei et al., 2024).

Table 1: Zero-shot One-to-One and Two-to-One retrieval accuracy. ($\mathcal{V}$: video, $\mathcal{A}$: audio, $\mathcal{T}$: text)

| One-to-One | | | | | Two-to-One | | | | |
| Method | Retrieval | Top-1 | Top-5 | Top-10 | Method | Retrieval | Top-1 | Top-5 | Top-10 |
|---|---|---|---|---|---|---|---|---|---|
| FABIND | $\mathcal{V} \to \mathcal{T}$ | 0.446 | 0.719 | 0.822 | FABind | $\mathcal{V}, \mathcal{A} \to \mathcal{T}$ | 0.309 | 0.665 | 0.781 |
| CENTROBIND | | **0.483** | **0.764** | **0.850** | | | | | |
| FABIND | $\mathcal{A} \to \mathcal{T}$ | 0.077 | 0.238 | 0.367 | CENTROBIND | | **0.745** | **0.957** | **0.978** |
| CENTROBIND | | **0.233** | **0.517** | **0.678** | | | | | |
| FABIND | $\mathcal{T} \to \mathcal{V}$ | **0.812** | **0.946** | **0.978** | FABIND | $\mathcal{T}, \mathcal{A} \to \mathcal{V}$ | 0.180 | 0.401 | 0.513 |
| CENTROBIND | | 0.591 | 0.839 | 0.909 | | | | | |
| FABIND | $\mathcal{A} \to \mathcal{V}$ | **0.058** | 0.154 | 0.226 | CENTROBIND | | **0.388** | **0.646** | **0.768** |
| CENTROBIND | | 0.052 | **0.184** | **0.284** | | | | | |
| FABIND | $\mathcal{T} \to \mathcal{A}$ | 0.201 | 0.438 | 0.584 | FABIND | $\mathcal{T}, \mathcal{V} \to \mathcal{A}$ | 0.099 | 0.257 | 0.364 |
| CENTROBIND | | **0.290** | **0.572** | **0.706** | | | | | |
| FABIND | $\mathcal{V} \to \mathcal{A}$ | 0.051 | 0.155 | 0.223 | CENTROBIND | | **0.232** | **0.490** | **0.625** |
| CENTROBIND | | **0.054** | **0.175** | **0.258** | | | | | |

## 4.1 EXPERIMENTS WITH SYNTHETIC DATASETS

We conduct controlled experiments on synthetic datasets generated from a Gaussian mixture model (Bishop & Nasrabadi, 2006). Appendix C.1 provides complete details on data generation, experimental settings, and results. These experiments evaluate whether CENTROBIND resolves the limitations in Section 2.3 across scenarios with varying numbers of modalities, modality imbalance, and encoder quality. Across all configurations, the synthetic results show that CENTROBIND overcomes the limitations of FABIND. Additional analyses (embedding visualizations, stability studies, temperature tuning, and complexity) are also included in Appendix C.1.

## 4.2 EXPERIMENTS WITH REAL-WORLD DATASETS

We evaluate and compare the performance of CENTROBIND and baseline methods on real-world datasets. See Appendix C.2 for detailed discussion on baselines and experimental results.

**Downstream tasks with MUStARD.** We perform evaluations in zero-shot binary and multi-class classification tasks, One-to-One, and Two-to-One cross-modal retrieval. For classification tasks, we use a Multi-Layer Perceptron (MLP) to perform sarcasm detection as a binary classification and speaker classification with 23 multi-class categories. In particular, MLP is trained on embeddings in a single modality (denoted by **Tr** in Table 2) and accuracy is evaluated on another modality (denoted by **Ev** in Table 2). In retrieval tasks, we measure the accuracy of correct retrieval. For the One-to-One case, we retrieve data sample in different modality by choosing the closest embedding from a single input embedding, while for the Two-to-One case we choose the closest embedding from the centroid of two input embeddings in two modalities. We denote input and target modalities with $\to$ in Table 1.

**Results on cross-modal retrieval.** Table 1 shows the performance for One-to-One and Two-to-One retrieval tasks. CENTROBIND consistently excels in One-to-One retrieval for text and audio modalities, while FABIND performs better for video retrieval. This might be due to the power of text to describe, which may be suitable for FABIND anchored at text modality. A notable observation is that the centroid of

Table 2: Zero-shot accuracy results. ($\mathcal{V}$: video, $\mathcal{A}$: audio, $\mathcal{T}$: text). Asterisks[*]: accuracy evaluated in different settings.

| Method | Tr, (Ev) | Sar-1 | Spk-1 | Spk-3 | Spk-5 |
|---|---|---|---|---|---|
| FABIND | $\mathcal{V}, (\mathcal{T})$ | 0.706 | 0.378 | 0.614 | 0.730 |
| UniBind | | 0.544 | 0.170 | 0.328 | 0.478 |
| AudioCLIP[*] | | 0.501 | 0.096 | 0.258 | 0.388 |
| ViT-Lens[*] | | 0.506 | 0.097 | 0.343 | 0.449 |
| CENTROBIND | | **0.716** | **0.474** | **0.736** | **0.836** |
| FABIND | $\mathcal{A}, (\mathcal{T})$ | 0.648 | 0.186 | 0.455 | 0.577 |
| UniBind | | 0.628 | 0.220 | 0.399 | 0.501 |
| AudioCLIP[*] | | 0.486 | 0.094 | 0.214 | 0.322 |
| ViT-Lens[*] | | 0.484 | 0.077 | 0.214 | 0.313 |
| CENTROBIND | | **0.691** | **0.290** | **0.546** | **0.714** |
| FABIND | $\mathcal{T}, (\mathcal{V})$ | 0.572 | 0.243 | 0.445 | 0.630 |
| UniBind | | 0.484 | 0.129 | 0.262 | 0.404 |
| AudioCLIP[*] | | 0.506 | 0.158 | 0.345 | 0.461 |
| ViT-Lens[*] | | 0.502 | 0.168 | 0.323 | 0.423 |
| CENTROBIND | | **0.694** | **0.368** | **0.670** | **0.791** |
| FABIND | $\mathcal{A}, (\mathcal{V})$ | 0.623 | 0.228 | **0.484** | 0.628 |
| UniBind | | 0.567 | 0.199 | 0.367 | 0.514 |
| AudioCLIP[*] | | 0.503 | 0.209 | 0.384 | 0.496 |
| ViT-Lens[*] | | 0.500 | 0.149 | 0.332 | 0.451 |
| CENTROBIND | | **0.683** | **0.243** | 0.475 | **0.632** |
| FABIND | $\mathcal{V}, (\mathcal{A})$ | 0.604 | 0.255 | 0.472 | 0.636 |
| UniBind | | 0.506 | 0.126 | 0.280 | 0.429 |
| AudioCLIP[*] | | 0.501 | 0.080 | 0.199 | 0.326 |
| ViT-Lens[*] | | 0.533 | 0.219 | 0.438 | 0.575 |
| CENTROBIND | | **0.626** | **0.326** | **0.548** | **0.703** |
| FABIND | $\mathcal{T}, (\mathcal{A})$ | 0.534 | 0.241 | 0.509 | 0.635 |
| UniBind | | 0.514 | 0.091 | 0.248 | 0.365 |
| AudioCLIP[*] | | 0.477 | 0.088 | 0.309 | 0.439 |
| ViT-Lens[*] | | 0.475 | 0.070 | 0.214 | 0.329 |
| CENTROBIND | | **0.655** | **0.346** | **0.610** | **0.741** |

video and audio embeddings achieves the best text retrieval performance. This implies complementary information exists and is captured by CENTROBIND.

**Results on sarcasm & speaker classification.** Table 2 presents results for sarcasm detection and speaker classification tasks, where Sar-1 indicates Top-1 accuracy for sarcasm, and Spk-$k$, $k = 1, 3, 5$ represent Top-$k$ accuracies for speaker classification. It is important to highlight that CENTROBIND and FABIND are trained on a single modality (**Tr**) and evaluated on a different modality (**Ev**) in a zero-shot setting, which can effectively measure the ability of multi-modal alignment.

In this experiment, CENTROBIND consistently outperforms FABIND and UniBind across all pairs of train and evaluation modalities, which can be distributed to CENTROBIND generally learning a better unified embedding space than FABIND. UniBind performs poorly in the zero-shot cross-modal experiment, which we believe is due to its insufficient multi-modal alignment. Since UniBind utilizes LLM-augmented descriptions for each modality and binds other encoders to these descriptions, multi-modal alignment may fail if the descriptions are dispersed across the embedding space. As analyzed in Section 2.3 and Section 3.2, these results highlight the CENTROBIND's ability to preserve intra and shared information among modalities, which are useful in unknown downstream tasks. Moreover, the zero-shot setting verifies the multi-modal alignment of CENTROBIND.

**Comparison with ImageBind backbone.** We compare the performance of FABIND and CENTROBIND using the ImageBind backbone (Girdhar et al., 2023) to assess whether CENTROBIND yields additional gains over ImageBind, which was pretrained on large-scale datasets and is optimized primarily for the image modality. As shown in Table 3, CENTROBIND outperforms ImageBind on the DreamBooth (Ruiz et al., 2023), AVE (Tian et al., 2018), AudioSet (Gemmeke et al., 2017), and UR-FUNNY (Hasan et al., 2019) datasets—even with 1) a strong pretrained backbone, 2) an image-anchored modality, and 3) a bimodal setup. We expect this gap to widen with a weaker backbone, without the image modality, or as additional modalities are introduced, as also evidenced by the synthetic and MUStARD results. These findings align with our analysis of dynamic anchor binding and highlight its effectiveness across nearly all evaluated scenarios. See Appendix C.3 for more discussion.

Table 3: Cross-modal retrieval accuracy. Performance dynamics are shown in Fig 11.

| | | IMAGEBIND | | CENTROBIND | |
|---|---|---|---|---|---|
| DATASET | MODALITIES | TOP1 | TOP5 | TOP1 | TOP5 |
| DREAMBOOTH | $\mathcal{V}, \mathcal{T}$ | 0.672 | 0.984 | **0.719** (↑ 0.047) | **1.000** (↑ 0.016) |
| AVE | $\mathcal{V}, \mathcal{A}$ | 0.313 | 0.649 | **0.327** (↑ 0.014) | **0.676** (↑ 0.027) |
| AUDIOSET | $\mathcal{V}, \mathcal{A}$ | 0.515 | 0.806 | **0.540** (↑ 0.025) | **0.824** (↑ 0.018) |
| UR-FUNNY | $\mathcal{V}, \mathcal{A}$ | 0.219 | 0.563 | **0.258** (↑ 0.039) | **0.598** (↑ 0.035) |
| UR-FUNNY | $\mathcal{V}, \mathcal{T}$ | 0.037 | **0.204** | **0.038** (↑ 0.001) | 0.202 (↓ 0.002) |
| DREAMBOOTH + AVE + AUDIOSET | $\mathcal{V}, \mathcal{T}, \mathcal{A}$ | 0.514 | 0.804 | **0.533** (↑ 0.019) | **0.816** (↑ 0.012) |

## 5 CONCLUSIONS

In this paper, we analyze the limitations of fixed-anchor-binding (FABIND) methods, including their over-reliance on a single anchor modality and their inability to capture both intra-modal and shared information among non-anchored modalities. To address these shortcomings, we propose adaptive anchor binding methods such as CENTROBIND, which align multi-modal embeddings to centroid-based anchors, removing the need for a fixed anchor modality. Our approach generalizes and extends methods like ImageBind. We also provide a theoretical analysis showing that CENTROBIND effectively captures both intra-modal and shared inter-modal information. Experiments on synthetic and real-world datasets demonstrate that CENTROBIND outperforms FABIND across nearly all settings—including modality imbalance, varying backbone quality, differing numbers of modalities (including bimodal cases), and a wide range of dataset sizes—yielding a robust, unified representation space and validating our theoretical insights.

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

# A   RELATED WORK

## A.1   MULTI-MODAL LEARNING

Multi-modal learning has gained significant attention in recent years due to its potential to enhance machine learning models by leveraging diverse data modalities, such as text, images, audio, and video. By combining these modalities, multi-modal learning seeks to mimic human-like perception, thereby improving performance across a wide range of applications, from healthcare to natural language processing. Common supervised multi-modal learning tasks include audio-visual classification (Peng et al., 2022; Feichtenhofer et al., 2019; Zhu & Rahtu, 2022), visual question answering (Antol et al., 2015; Guo et al., 2021), and vision-language tasks (Xu et al., 2015; Radford et al., 2021), as well as vision-audio-language tasks (Aytar et al., 2017; Harwath et al., 2018).

Typically, these models integrate uni-modal features extracted by modality-specific encoders (Seichter et al., 2021; Nagrani et al., 2021; Wu et al., 2022; Wang et al., 2020; Peng et al., 2022). For instance, (Madaan et al., 2024) introduces inter- and intra-modality modeling frameworks that treat the target as a composition of multiple modalities. Similarly, (Du et al., 2023) proposes a late-fusion approach for supervised multi-modal tasks, demonstrating that insufficient feature extraction from individual modalities negatively affects the model's generalization ability. Additionally, (Zhang et al., 2024) addresses joint optimization by alternating between uni-modal learning scenarios and integrating modality-specific encoders with a unified head shared across all modalities.

Multi-view learning is closely related to multi-modal learning. Early work in (Tian et al., 2020a) studies multi-view representation learning through predictive and contrastive approaches across multiple modalities, demonstrating the efficacy of contrastive learning in multi-view settings. Building on these insights, numerous subsequent studies have integrated contrastive learning into multi-view tasks. For example, (Xing et al., 2019) proposes an approach that adaptively combines image-text representations for few-shot learning; however, this method still relies on labeled data. Some research leverages anchors or a fused modality as the anchor representation. (Zeng et al., 2021) introduces a unified prototype representation to address cross-modal retrieval imbalance by employing a reconstruction loss, but its reliance on unified prototypes as anchors restricts the capture of inter-view information across modalities. Meanwhile, (Huang et al., 2023) develops a tri-modal alignment pre-training task—extending text-video alignment to include a fused modality using pairwise contrastive learning; however, it does not explicitly handle intra-view learning. Furthermore, (Wang et al., 2022b) presents a cross-modal data augmentation technique for image-text multi-view learning, randomly replacing visually grounded words with diverse image patches to increase data variety and encourage token-level interaction across modalities. (Dufumier et al., 2024) introduces a contrastive multimodal approach that maximizes mutual information between augmented multi-modal features by effectively capturing redundant, unique, and synergistic interactions across modalities beyond traditional multi-modal alignment constraints. (Lin et al.) presents inter-modality alignment combined with boundary expansion for multi-view classification to mitigate information redundancy. Nevertheless, this approach still overlooks intra-view information, indicating the need for methods that jointly consider both inter- and intra-view representations.

## A.2   MULTI-MODAL ALIGNMENT

Multi-modal learning addresses four key challenges (Liang et al., 2024c; Baltrušaitis et al., 2018; Liang et al., 2024d): managing interactions among redundant, unique, and synergistic features (Dumas et al., 2017; Liang et al., 2024a;b), aligning fine-grained and coarse-grained information (Wang et al., 2023; 2024a), reasoning across diverse features (Yang et al., 2023), and integrating external knowledge (Shen et al., 2022; Lyu et al., 2024). Among these challenges, multi-modal alignment is one of the core challenges that many researchers aim to solve.

A common method in multi-modal alignment is using cross-modal alignment by using attention mechanisms between pairwise modalities, such as vision-language (Tan & Bansal, 2019) and vision-language-audio (Tsai et al., 2019). Another effective approach is leveraging graph neural networks to align multi-modal datasets (Yang et al., 2021; Wilf et al., 2023). For instance, (Yang et al., 2021) transforms unaligned multi-modal sequence data into nodes, with edges capturing interactions across modalities over time. (Wilf et al., 2023) builds graph structures for each modality—visual, textual, and acoustic—and create edges to represent their interactions.

To enhance the generalizability of cross-modal representations, (Xia et al., 2024) employs a unified codebook approach, facilitating a joint embedding space for visual and audio modalities. Another prominent method achieves cross-modal alignment by leveraging large collections of image-text pairs, making it a widely adopted strategy in multi-modal learning (Radford et al., 2021; Zhang et al., 2022; Guzhov et al., 2022; Zhou et al., 2023).

### A.3 MULTI-MODAL DOMAIN GENERALIZATION

Multimodal domain generalization (MMDG) aims to train models on data from multiple modalities (e.g., image, audio, text) and source domains, such that the models generalize to unseen target domains that share the same modalities. A primary challenge in MMDG lies in aligning heterogeneous modality-specific representations into a shared embedding space while preserving both shared semantics and modality-specific information.

(Fan et al., 2024) reduces generalization error by flattening the representation-space loss landscape using multi-modal feature interpolation and teacher-student distillation. This strategy mitigates modality dominance, in which stronger modalities (e.g., images) overpower weaker ones (e.g., audio), but it aligns representations only between modality pairs and assumes that all modalities are present during both training and inference. (Dong et al., 2023) decomposes modality features into shared and specific components and apply supervised contrastive learning to the shared part using label supervision. They further use cross-modal translation modules to reconstruct one modality's representation from another. However, their framework depends on modality-specific translation paths and requires prior knowledge of available modalities, limiting its scalability to new combinations. (Dong et al., 2024) extends the setting to open-set domain generalization by introducing self-supervised tasks, masked cross-modal translation and multi-modal jigsaw puzzles, that enhance the model's ability to detect unknown classes. While effective for class-level generalization, their method assumes a fixed modality set and does not support unseen modalities at test time.

In contrast, CENTROBIND constructs a unified and modality-agnostic embedding space by dynamically computing centroid-based anchors from all available modality embeddings within a sample. Instead of aligning only modality pairs or relying on fixed translation modules, CENTROBIND pulls each modality's representation toward the centroid, enabling joint alignment without architectural changes. This approach supports any combination of available or unseen modalities and eliminates the need for predefined modality decomposition, making CENTROBIND scalable and adaptive for real-world multimodal scenarios.

### A.4 BINDING METHODS

Recent studies have focused on aligning multi-modal datasets by leveraging binding properties in various modalities. ImageBind (Girdhar et al., 2023) align multi-modal data by using image representation as the anchor and aligning each modality embedding with the image embedding. Similarly, LanguageBind (Zhu et al., 2024) use language representation as the anchor, aligning other modalities into the language space. PointBind (Guo et al., 2023) learn a joint embedding space across 3d point, language, image, and audio modalities by designating the point space as the central representation. Thanks to the efficacy of such a binding idea with a fixed anchor, several "-Bind" approaches have been studied in numerous domains (Teng et al., 2024; Xiao et al., 2024; Gao et al., 2024; Yang et al., 2024b; Balemans et al., 2024; Dhakal et al., 2024; Yang et al., 2024a) While these methods demonstrate strong performance in zero-shot cross-modality retrieval and classification tasks, they are constrained by their reliance on an existing single anchor modality.

Several approaches have integrated additional knowledge into multi-modal representation spaces to address this limitation. Freebind (Wang et al., 2024a) introduce bi-modality spaces to enhance a pretrained image-paired unified space. It generates pseudo-embedding pairs across diverse modality pairs and aligns them with the pre-trained unified space using contrastive learning. Omnibind (Wang et al., 2024b) leverage multiple pretrained multi-modal models to construct pseudo item-pair retrievals based on top-1 recall across various modality combinations using pairwise cross-modal alignment. Both methods show promising results in cross-modal retrieval by incorporating extra spaces into existing pairwise binding spaces. However, they still rely on fixed (pre-trained) representation spaces.

Unibind (Lyu et al., 2024) highlight the imbalanced representation when using image-centered representation spaces. To address this, Unibind employs large language models (LLMs) to create a unified and balanced representation space. It constructs a knowledge base with multi-modal category descriptions, establishes LLM-augmented class-wise embedding centers, and aligns other modalities to these centers through contrastive learning. This approach attempts to balance representations across modalities but still depends heavily on large-scale pretrained LLMs and centers alignment around a single unified space, namely, text (language).

ViT-Lens (Lei et al., 2024) build upon the Vision Transformer (ViT) (Dosovitskiy et al., 2021) and multi-modal foundational models like CLIP (Radford et al., 2021) to align multiple modalities. It extends ViT by incorporating an additional embedding layer and attention layer for each modality, which are trained via contrastive learning involving embeddings generated by the CLIP and the ViT models. This approach generalizes FABIND by allowing more than one fixed anchor modality; specifically, image and text in this case. CENTROBIND could also adopt a similar strategy, leveraging the powerful ViT model for modality alignment while adaptively computing anchors based on their centroids.

## B PROOFS

### B.1 PROOF OF PROPOSITION 2.2

Using the chain rule of the mutual information, we observe that

$$
\begin{aligned}
I(\mathbf{X}_1, f_1^{\mathrm{suf}}(\mathbf{X}_1); \mathbf{X}_i) &= I(\mathbf{X}_1; \mathbf{X}_i) + I(f_1^{\mathrm{suf}}(\mathbf{X}_1); \mathbf{X}_i|\mathbf{X}_1) \\
&= I(f_1^{\mathrm{suf}}(\mathbf{X}_1); \mathbf{X}_i) + I(\mathbf{X}_1; \mathbf{X}_i|f_1^{\mathrm{suf}}(\mathbf{X}_1)),
\end{aligned}
\tag{12}
$$

Since $f_1^{\mathrm{suf}}(\mathbf{X}_1)$ is a deterministic function of $\mathbf{X}_1$, we have

$$
I(f_1^{\mathrm{suf}}(\mathbf{X}_1); \mathbf{X}_i|\mathbf{X}_1) = 0.
\tag{13}
$$

Moreover, $f_1^{\mathrm{suf}}$ obtained in Definition 2.1 with proper choice of $\mathcal{Z}$ achieves the maximum mutual information, implying together with $I(\mathbf{X};\mathbf{Y}) \leq \min\{H(\mathbf{X}), H(\mathbf{Y})\}$ that $I(f_1^{\mathrm{suf}}(\mathbf{X}_1); \mathbf{X}_1) = H(\mathbf{X}_1)$, where $H(\mathbf{X}_1)$ is the entropy of $\mathbf{X}_1$ (Polyanskiy & Wu, 2024). In other words, we have $H(\mathbf{X}_1|f_1^{\mathrm{suf}}(\mathbf{X}_1)) = H(\mathbf{X}_1) - I(f_1^{\mathrm{suf}}(\mathbf{X}_1); \mathbf{X}_1) = 0$. This gives

$$
\begin{aligned}
I(\mathbf{X}_1; \mathbf{X}_i|f_1^{\mathrm{suf}}(\mathbf{X}_1)) &= H(\mathbf{X}_1|f_1^{\mathrm{suf}}(\mathbf{X}_1)) - H(\mathbf{X}_1|f_1^{\mathrm{suf}}(\mathbf{X}_1), \mathbf{X}_i) \\
&= 0
\end{aligned}
\tag{14}
$$

Substituting (13) and (14) into (12) yields

$$
I(\mathbf{X}_1; \mathbf{X}_i) = I(f_1^{\mathrm{suf}}(\mathbf{X}_1); \mathbf{X}_i).
\tag{15}
$$

We conclude the proof of Proposition 2.2 by noting that the optimality of FABIND (i.e., $I(f_1^{\mathrm{suf}}(\mathbf{X}_1); \mathbf{X}_i) = I(f_1^{\mathrm{suf}}(\mathbf{X}_1); f_i^{\mathrm{FB}}(\mathbf{X}_i))$, $\forall i \in \{2, \cdots, M\}$) yields

$$
I(\mathbf{X}_1; \mathbf{X}_i) = I(f_1^{\mathrm{suf}}(\mathbf{X}_1); f_i^{\mathrm{FB}}(\mathbf{X}_i)).
\tag{16}
$$

### B.2 PROOF OF PROPOSITION 2.3

Using the chain rule of mutual information, we have

$$
\begin{aligned}
I(f_1^{\mathrm{ins}}(\mathbf{X}_1); \mathbf{X}_1, \mathbf{X}_i) &= I(f_1^{\mathrm{ins}}(\mathbf{X}_1); \mathbf{X}_1) + I(f_1^{\mathrm{ins}}(\mathbf{X}_1); \mathbf{X}_i|\mathbf{X}_1) \\
&= I(f_1^{\mathrm{ins}}(\mathbf{X}_1); \mathbf{X}_i) + I(f_1^{\mathrm{ins}}(\mathbf{X}_1); \mathbf{X}_1|\mathbf{X}_i).
\end{aligned}
\tag{17}
$$

Moreover, since $f_1^{\mathrm{ins}}(\mathbf{X}_1)$ is a deterministic function of $\mathbf{X}_1$, we have $I(f_1^{\mathrm{ins}}(\mathbf{X}_1); \mathbf{X}_i|\mathbf{X}_1) = 0$, leading to $I(f_1^{\mathrm{ins}}(\mathbf{X}_1); \mathbf{X}_1) = I(f_1^{\mathrm{ins}}(\mathbf{X}_1); \mathbf{X}_i) + I(f_1^{\mathrm{ins}}(\mathbf{X}_1); \mathbf{X}_1|\mathbf{X}_i)$. Then, using the assumption $I(f_1^{\mathrm{ins}}(\mathbf{X}_1); \mathbf{X}_1) < \epsilon$, it follows that

$$
\begin{aligned}
\epsilon &> I(f_1^{\mathrm{ins}}(\mathbf{X}_1); \mathbf{X}_i) + I(f_1^{\mathrm{ins}}(\mathbf{X}_1); \mathbf{X}_1|\mathbf{X}_i) \\
&\overset{(a)}{\geq} I(f_1^{\mathrm{ins}}(\mathbf{X}_1); \mathbf{X}_i) \\
&\overset{(b)}{\geq} I(f_1^{\mathrm{ins}}(\mathbf{X}_1); f_i^{\mathrm{FB}}(\mathbf{X}_i)),
\end{aligned}
\tag{18}
$$

where the labeled inequalities follow from: (a) the non-negativity of mutual information; (b) the data processing inequality. This concludes the proof of Proposition 2.3.

### B.3 PROOF OF THEOREM 3.1

To prove Theorem 3.1, we leverage the reverse inequality of $M$-variable Hölder inequality (Seo, 2013, eq. (2.8)). For the sake of completeness, we state the inequality in Lemma B.1.

**Lemma B.1** (Reverse inequality of the $M$-variable Hölder inequality (Seo, 2013)). *Consider $M$ sequences $(x_{i,j})_{j \in [n]}$, $i \in [M]$ of $n$ positive scalars such that for some $0 < c_m \le c_M < \infty$,*

$$0 < c_m \le x_{i,j} \le c_M < \infty, \ \forall i, j. \tag{19}$$

*Then,*

$$\prod_{i=1}^{M} \left( \sum_{j=1}^{n} x_{i,j} \right)^{\frac{1}{n}} \le \frac{(c_m + c_M)^2}{4 c_m c_M} \sum_{j=1}^{n} \left( \prod_{i=1}^{M} x_{i,j} \right)^{\frac{1}{n}}. \tag{20}$$

Now we start by writing the summation of InfoNCE losses for each $f_l^{(t)}(\boldsymbol{x}'_{l,k}), l \in [M]$ to $f_i(\mathbf{X}_i)$ as

$$\sum_{l=1}^{M} I_{\mathrm{NCE}}(f_l(\mathbf{X}'_l); f_i(\mathbf{X}_i)|\tau) = -\frac{1}{|\mathcal{I}_B|} \sum_{k=1}^{|\mathcal{I}_B|} \sum_{l=1}^{M} \log \frac{\exp\left(\frac{f_l^\top(\boldsymbol{x}'_{l,k}) f_i(\boldsymbol{x}_{i,k})}{\tau}\right)}{\sum_{j \in \mathcal{I}_B} \exp\left(\frac{f_l^\top(\boldsymbol{x}'_{l,k}) f_i(\boldsymbol{x}_{i,j})}{\tau}\right)}. \tag{21}$$

Then, the inner summation in (21) is bounded as

$$\sum_{l=1}^{M} \log \frac{\exp\left(\frac{f_l^\top(\boldsymbol{x}'_{l,k}) f_i(\boldsymbol{x}_{i,k})}{\tau}\right)}{\sum_{j \in \mathcal{I}_B} \exp\left(\frac{f_l^\top(\boldsymbol{x}'_{l,k}) f_i(\boldsymbol{x}_{i,j})}{\tau}\right)}$$

$$= \frac{1}{\tau} \sum_{l=1}^{M} f_l^\top(\boldsymbol{x}'_{l,k}) f_i(\boldsymbol{x}_{i,k}) - \log \prod_{l=1}^{M} \sum_{j \in \mathcal{I}_B} \exp\left(\frac{f_l^\top(\boldsymbol{x}'_{l,k}) f_i(\boldsymbol{x}_{i,j})}{\tau}\right)$$

$$\overset{(a)}{\ge} \frac{1}{\tau} \sum_{l=1}^{M} f_l^\top(\boldsymbol{x}'_{l,k}) f_i(\boldsymbol{x}_{i,k}) - \log \left( C_{\mathcal{F},k,i} \sum_{j \in \mathcal{I}_B} \prod_{l=1}^{M} \exp\left(\frac{f_l^\top(\boldsymbol{x}'_{l,k}) f_i(\boldsymbol{x}_{i,j})}{\tau |\mathcal{I}_B|}\right) \right)^{|\mathcal{I}_B|}$$

$$\overset{(b)}{=} \frac{M}{\tau} \boldsymbol{a}_k^\top f_i(\boldsymbol{x}_{i,k}) - |\mathcal{I}_B| \log \sum_{j \in \mathcal{I}_B} \exp\left(\frac{M \boldsymbol{a}_k^\top f_i(\boldsymbol{x}_{i,j})}{\tau |\mathcal{I}_B|}\right) - |\mathcal{I}_B| \log C_{\mathcal{F},k,i}$$

$$= |\mathcal{I}_B| \log \exp\left(\frac{M \boldsymbol{a}_k^\top f_i(\boldsymbol{x}_{i,k})}{\tau |\mathcal{I}_B|}\right) - |\mathcal{I}_B| \log \sum_{j \in \mathcal{I}_B} \exp\left(\frac{M \boldsymbol{a}_k^\top f_i(\boldsymbol{x}_{i,j})}{\tau |\mathcal{I}_B|}\right) - |\mathcal{I}_B| \log C_{\mathcal{F},k,i}$$

$$= |\mathcal{I}_B| \log \frac{\exp\left(\frac{M \boldsymbol{a}_k^\top f_i(\boldsymbol{x}_{i,k})}{\tau |\mathcal{I}_B|}\right)}{\sum_{j \in \mathcal{I}_B} \exp\left(\frac{M \boldsymbol{a}_k^\top f_i(\boldsymbol{x}_{i,j})}{\tau |\mathcal{I}_B|}\right)} - |\mathcal{I}_B| \log C_{\mathcal{F},k,i}, \tag{22}$$

where the labeled (in)equalities follow from: (a) Lemma B.1 and $C_{\mathcal{F},k,i} = \frac{(c_{\mathcal{F},k,i}^{\min} + c_{\mathcal{F},k,i}^{\max})^2}{4 c_{\mathcal{F},k,i}^{\min} c_{\mathcal{F},k,i}^{\max}}$ with

$$c_{\mathcal{F},k,i}^{\min} = \min_{\ell \in [M], j \in \mathcal{I}_B} \exp\left(\frac{f_l^\top(\boldsymbol{x}'_{l,k}) f_i(\boldsymbol{x}_{i,j})}{\tau}\right), \text{ and}$$

$$c_{\mathcal{F},k,i}^{\max} = \max_{\ell \in [M], j \in \mathcal{I}_B} \exp\left(\frac{f_l^\top(\boldsymbol{x}'_{l,k}) f_i(\boldsymbol{x}_{i,j})}{\tau}\right); \tag{23}$$

and (b) the definition of anchor embedding (7). Substituting (22) into (21) gives

$$\sum_{l=1}^{M} I_{\mathrm{NCE}}(f_l(\mathbf{X}'_l); f_i(\mathbf{X}_i)|\tau) \le -\frac{1}{|\mathcal{I}_B|} \sum_{k=1}^{|\mathcal{I}_B|} \left[ |\mathcal{I}_B| \log \frac{\exp\left(\frac{M \boldsymbol{a}_k^\top f_i(\boldsymbol{x}_{i,k})}{\tau |\mathcal{I}_B|}\right)}{\sum_{j \in \mathcal{I}_B} \exp\left(\frac{M \boldsymbol{a}_k^\top f_i(\boldsymbol{x}_{i,j})}{\tau |\mathcal{I}_B|}\right)} - |\mathcal{I}_B| \log C_{\mathcal{F},k,i} \right]$$

$$= |\mathcal{I}_B| I_{\mathrm{NCE}}\left( \mathbf{A}; f_i(\mathbf{X}_i) \ \middle| \ \frac{\tau |\mathcal{I}_B|}{M} \right) + \sum_{k=1}^{|\mathcal{I}_B|} \log C_{\mathcal{F},k,i}. \tag{24}$$

Rearranging (24) and setting $\tilde{\tau} = \frac{\tau|\mathcal{I}_B|}{M}$ in (23) and (24) yield

$$I_{\text{NCE}}\left(\mathbf{A}; f_i(\mathbf{X}_i) \mid \tilde{\tau}\right) \geq \frac{1}{|\mathcal{I}_B|} \sum_{l=1}^{M} I_{\text{NCE}}\left(f_l(\mathbf{X}_l'); f_i(\mathbf{X}_i) \,\Big|\, \frac{\tilde{\tau}M}{|\mathcal{I}_B|}\right) - \frac{1}{|\mathcal{I}_B|} \sum_{k=1}^{|\mathcal{I}_B|} \log C_{\mathcal{F},k,i}, \quad (25)$$

which concludes the proof of Theorem 3.1.

## C  EXPERIMENTS

---
**Algorithm 1** CENTROBIND

---
Initialize encoders $f_1^{(0)}, f_2^{(0)}, \cdots, f_M^{(0)}$.
**for** $t = 0, 1, \ldots, t_{\max}$ **do**
    Sample $B$ from multi-modal datasets $\{\mathcal{D}_i\}_i$.
    Generate anchor embeddings $\{\boldsymbol{a}_j\}_{j \in \mathcal{I}_B}$ using (7)
    **for** $i = 1, \ldots, M$ **do**
        Minimize $\mathcal{L}_{\text{CB}}(f_i^{(t+1)}|\tau)$ in (8)
    **end for**
**end for**

---

In this section, we provide experimental details and additional results. Algorithmic expression of CENTROBIND is given in Algorithm 1.

### C.1  EXPERIMENTS WITH SYNTHETIC DATASETS

**Synthetic datasets.**  We employ a latent variable model (Bishop & Nasrabadi, 2006) for generating synthetic multi-modal datasets. A latent variable model is a statistical model for data $\mathbf{X} \in \mathbb{R}^{d_x}$, under which $\mathbf{X}$ is generated according to a conditional probability distribution $P_{\mathbf{X}|\mathbf{Z}}$, where $\mathbf{Z} \in \mathbb{R}^{d_z}$ is the latent variable. In terms of the representation learning framework, $\mathbf{Z}$ can be seen as a low dimensional representation of $\mathbf{X}$. We assume that the class label $\mathbf{Y} \in [K]$ and the latent variable $\mathbf{Z}$ are jointly distributed according to $P_{\mathbf{Z},\mathbf{Y}}$. In our setting, we exploit Gaussian mixture model (GMM) (Bishop & Nasrabadi, 2006) for the latent variable $\mathbf{Z}$, and we generate $M$ modalities $\mathbf{X}_i = g_i(\mathbf{Z}) + \mathbf{N}, i \in [M]$ with random noise $\mathbf{N}$ and some non-linear projections $g_i : \mathbb{R}^{d_z} \to \mathbb{R}^{d_x}$. We choose the projections in a way such that each model can be ranked in ascending order, i.e., $\mathbf{X}_1$ is the worst, and $\mathbf{X}_4$ is the best modality in terms of their inherent correlation with the latent variable. The class label $\mathbf{Y}$ is set to the component id of GMM.

In particular, the PDF of $\mathbf{Z}$ is defined as follows:

$$p_{\mathbf{Z}}(\boldsymbol{z}) = \prod_{y=1}^{K} \pi_y \mathcal{N}(\boldsymbol{z}; \boldsymbol{\mu}_y, \boldsymbol{\Sigma}_y), \quad (26)$$

where $K$ is the number of mixture components, $\pi_y = \Pr(\mathbf{Y} = y)$ is the component prior probability, and $\mathcal{N}(\boldsymbol{z}; \boldsymbol{\mu}_y, \boldsymbol{\Sigma}_y)$ denotes Gaussian PDF with mean $\boldsymbol{\mu}_y \in \mathbb{R}^{d_z}$ and covariance matrix $\boldsymbol{\Sigma}_y \in \mathbb{R}^{d_z \times d_z}$. This leads to the conditional PDF of $\mathbf{Z}$ as $p_{\mathbf{Z}|\mathbf{Y}}(\boldsymbol{z}|y) = \mathcal{N}(\boldsymbol{z}; \boldsymbol{\mu}_y, \boldsymbol{\Sigma}_y)$.

Once a latent variable $\boldsymbol{z}$ is generated from GMM in (26), we generate data samples $(\boldsymbol{x}_{i,1}, \boldsymbol{x}_{i,2}, \cdots, \boldsymbol{x}_{i,N})$ for $i$-th modality using the conditional PDFs of $\mathbf{X}_i$ given $\boldsymbol{z}$, denoted by $p_{\mathbf{X}_i|\mathbf{Z}}(\boldsymbol{x}_i|\boldsymbol{z})$. Specifically, we use the model $\mathbf{X}_i = g_i(\mathbf{Z}_i) + \mathbf{N}$, where $g_i : \mathbb{R}^{d_z} \to \mathbb{R}^{d_x}$ is a non-linear projection from latent space to observation space, and $\mathbf{N} \sim \mathcal{N}(\mathbf{0}, I_{d_x})$ is Gaussian noise with zero-mean and identity covariance matrix. To make the inherent correlation between $\mathbf{X}_i$ and $\mathbf{Z}_i$ different among modalities, we choose $g_i$ such that

$$g_i(\mathbf{Z}) = \Theta_i^{(2)} \text{sigmoid}\left(\Theta_i^{(1)} \mathbf{Z}\right), \quad (27)$$

where $\text{sigmoid}(x) = \frac{1}{1+e^{-x}}$ is applied element-wise, and $\Theta_i^{(1)} \in \mathbb{R}^{d_x \times d_z}$ and $\Theta_i^{(2)} \in \mathbb{R}^{d_x \times d_x}$ are matrices randomly generated from Gaussian distribution. Moreover, after $\Theta_i^{(1)}, i \in [M]$ are

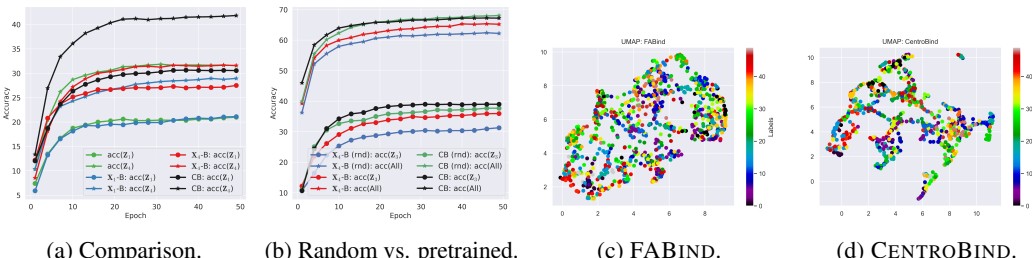

(a) Comparison.  (b) Random vs. pretrained.  (c) FABIND.  (d) CENTROBIND.

Figure 3: (a) and (b): Accuracy as a measure of the representation space quality. Abbreviation: $\mathbf{X}_i$-B or CB: applying FABIND with anchor $\mathbf{X}_i$ or applying CentroBind; acc($\mathbf{Z}_i$) or acc(All): accuracy of $\mathbf{Z}_i$ or of concatenated embeddings $(\mathbf{Z}_1, \cdots, \mathbf{Z}_M)$; (rnd): if random backbones are used. (c) and (d): Representation visualization via UMAP.

generated, we set arbitrary columns of them all zero, so that the number of all zero columns decreases in $i$. For example, 60% of columns of $\Theta_1^{(1)}$ are all-zero, while only 10% of columns of $\Theta_M^{(1)}$ are all-zero. This enables approximate control the correlation between $\mathbf{X}_i$ and $\mathbf{Z}$, providing estimates of best modality ($\mathbf{X}_M$) or worst modality ($\mathbf{X}_1$). To have meaningful labels for this latent model, which requires for downstream tasks, we set the labels $\mathbf{Y}$ being the component index in GMM. In particular, since there are $K$ components in GMM (26), there exist $K$ categories in $\mathbf{Y}$. We conduct experiments with three different synthetic datasets by setting $M = 4, 6, 8$. For all synthetic datasets, we fix $d_x = 16$, $d_z = 8$, and $K = 50$.

**Experiment details.** We initialize two different versions of backbones for all modalities, where the first is a random backbone (highlighted by (rnd) in figures), and the second is a backbone pretrained with InfoNCE loss. For each backbone, we use a simple multilayer perceptron (MLP). Comparing the results with these two versions of backbone provides how much both FABIND and CENTROBIND are robust to backbone quality. Given the backbones for $M$ modalities, we align the corresponding embedding spaces using either FABIND with anchor $\mathbf{X}_i$ (denoted by $\mathbf{X}_i$-B in figures) or CENTROBIND (denoted by CB in figures). Finally, with the encoders aligned by either FABIND or CENTROBIND, we evaluate classification accuracy as a measure of representation quality. We use a simple MLP for the classifier. To distinguish between accuracy with embeddings from a single modality and the one with concatenated embeddings from all modalities, we denote by acc($\mathbf{Z}_i$) the accuracy with embeddings from $i$-th modality and by acc(All) the accuracy with embeddings from all modalities. Specifically, for acc(All), we fuse the multi-modal embeddings using MLP layers. Therefore, the accuracy of the multi-modal case without binding methods (e.g., $\times$ method and the multi-modal column in Table 4) can be considered a naive baseline for multi-modal learning.

**Comparison with baseline methods.** Figure 3 shows the validation accuracy of each method (without binding, FABIND with anchor $\mathbf{X}_1$, FABIND with anchor $\mathbf{X}_4$, and CENTROBIND). For the same experimental setting, Figure 4 includes additional accuracy curves for $acc(\mathbf{Z}_1)$ and $acc(All)$. For better readability, the corresponding accuracy is provided in Table 4.

We conduct experiments with two types of backbone encoders: randomly initialized backbones and pre-trained backbones. For each type, we extract embeddings using four different methods: representations without binding (denoted by $\times$ in Table 5), FABIND with anchor modality $\mathbf{X}_1$ (denoted as FABIND-$\mathbf{X}_1$), FABIND with anchor modality $\mathbf{X}_4$ (denoted as FABIND-$\mathbf{X}_4$), and CENTROBIND. The embedding quality is then evaluated using classification accuracy. Specifically, we train five different decoders for each case: four uni-modal decoders (one for each modality) and one multi-modal decoder for the concatenated embeddings of all modalities. The results show that CENTROBIND outperforms the other baseline methods. Notably, CENTROBIND demonstrates superior performance in the case of randomly initialized backbones, indicating robustness to poor backbone quality.

Additional experimental results on synthetic datasets with $M = 6$ and $M = 8$ modalities are presented in Figure 8 and Figure 9, respectively. These results exhibit similar trends to those observed

Table 4: Classification accuracies presented in Figure 3.

| Backbone | Method | uni-modal | | | | multi-modal |
| --- | --- | --- | --- | --- | --- | --- |
| | | $\mathbf{X}_1$ | $\mathbf{X}_2$ | $\mathbf{X}_3$ | $\mathbf{X}_4$ | $\mathbf{X}_1, \cdots, \mathbf{X}_4$ |
| Pre-trained | $\times$ | 0.2166 | 0.2878 | 0.3536 | 0.3923 | 0.6985 |
| | FABIND-$\mathbf{X}_1$ | 0.2180 | 0.2736 | 0.3210 | 0.2999 | 0.5541 |
| | FABIND-$\mathbf{X}_4$ | 0.2483 | 0.3349 | **0.4207** | 0.3896 | **0.7024** |
| | CENTROBIND | **0.2540** | **0.3433** | 0.4162 | **0.4559** | 0.6974 |
| Random | $\times$ | 0.2109 | 0.2472 | 0.2597 | 0.2815 | 0.6648 |
| | FABIND-$\mathbf{X}_1$ | 0.2119 | 0.2587 | 0.3034 | 0.3081 | 0.5502 |
| | FABIND-$\mathbf{X}_4$ | 0.2447 | 0.3076 | 0.3826 | 0.2813 | 0.6742 |
| | CENTROBIND | **0.2582** | **0.3392** | **0.4224** | **0.4649** | **0.7006** |

with $M = 4$ modalities. These experiments verify that CENTROBIND is capable of handling a large number of modalities effectively.

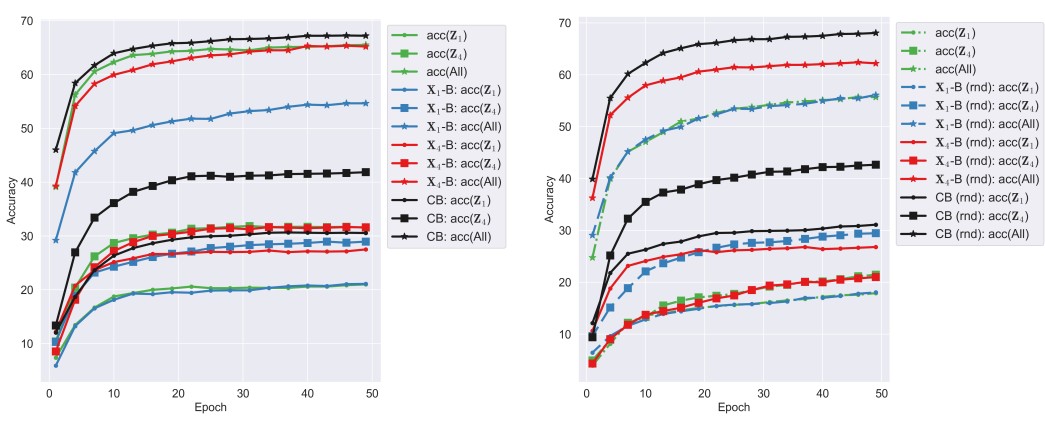

(a) Pre-trained backbones are used.     (b) Randomly initialized backbones are used.

Figure 4: Accuracy as a measure of the representation space quality. Abbreviation: $\mathbf{X}_i$-B or CB: applying FABIND with anchor $\mathbf{X}_i$ or applying CentroBind; acc($\mathbf{Z}_i$) or acc(All): accuracy of $\mathbf{Z}_i$ or of concatenated embeddings $(\mathbf{Z}_1, \cdots, \mathbf{Z}_M)$; (rnd): if random backbones are used.

**Representation visualization.**  Figure 5 presents t-SNE (Van der Maaten & Hinton, 2008) and UMAP (McInnes et al., 2018) visualizations of embeddings generated by FABIND and CENTROBIND. For this visualization, we use synthetic datasets with 4 modalities, ensuring that each modality is equally informative, and plot the embeddings for $\mathbf{X}_1$. FABIND is anchored at $\mathbf{X}_4$, and both binding methods utilize pre-trained backbones.

In both t-SNE and UMAP visualizations, CENTROBIND produces more clustered representations, whereas FABIND results in more scattered embeddings. These findings validate our analysis that CENTROBIND creates a superior representation space by effectively learning both intra and shared information.

**Convergence and stability analysis.**  The convergence rate of CENTROBIND may differ from that of FABIND due to the replacement of the fixed anchor with a dynamic anchor. In Figure 6, we plot the loss curves of CENTROBIND and FABIND during training. The results show that the loss of CENTROBIND saturates earlier than that of FABIND. We attribute this to the fact that the centroid serves as a minimizer of embeddings in terms of Euclidean distance, making it easier to converge embeddings to their centroid compared to converging them to one specific embedding.

The plot also reveals a crossover point where the loss curves intersect. We believe this occurs due to the number of InfoNCE losses optimized by CENTROBIND and FABIND. Specifically, with $M$ modalities, CENTROBIND minimizes $M$ InfoNCE losses, while FABIND minimizes $M - 1$ InfoNCE

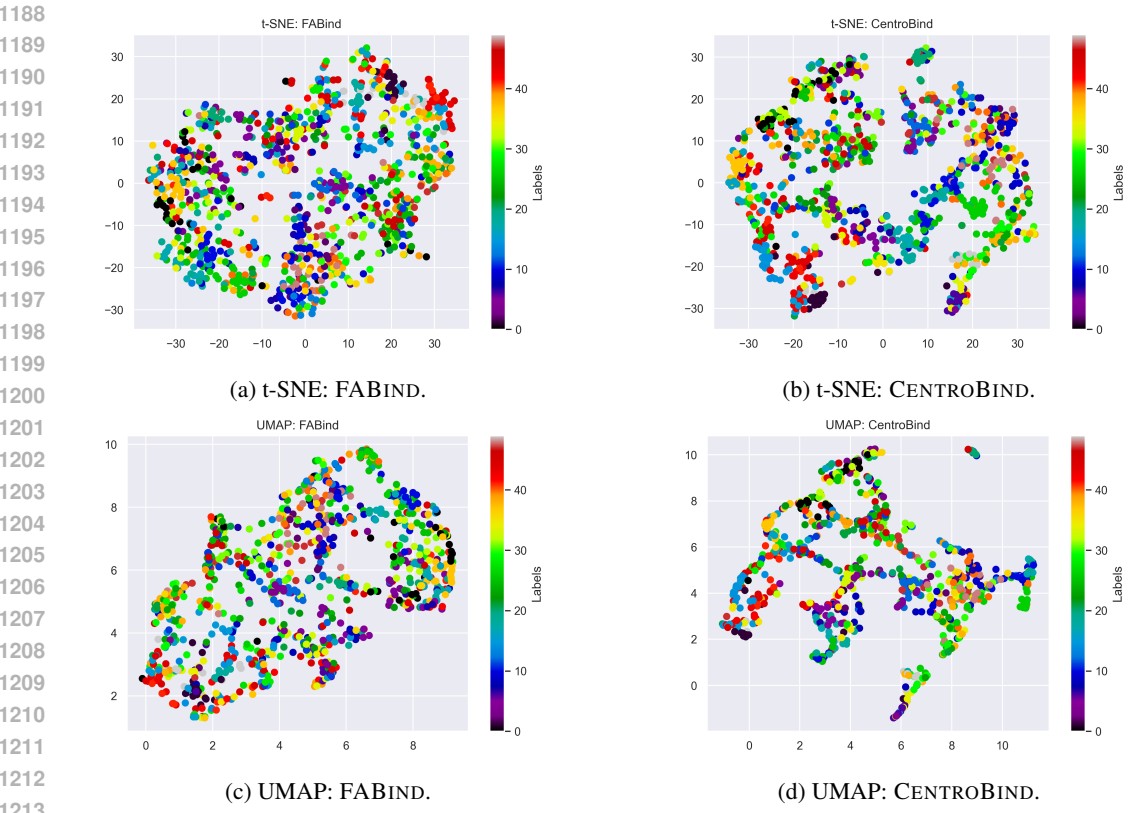

(a) t-SNE: FABIND.

(b) t-SNE: CENTROBIND.

(c) UMAP: FABIND.

(d) UMAP: CENTROBIND.

Figure 5: Representation visualization via t-SNE and UMAP.

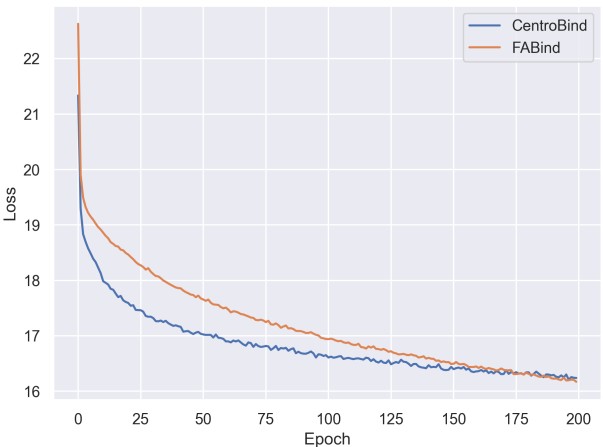

Figure 6: Training loss.

losses. This results in a smaller loss for FABIND when the encoders are well-trained, which explains the crossover point observed in Figure 6.

**Temperature sensitivity.** We examine the effect of temperature parameter $\tau$ in InfoNCE loss by evaluating classification accuracy on the GMM synthetic dataset with different $\tau \in \{0.07, 0.3, 0.7\}$. Figure 7 displays the classification accuracy for each temperature setting. Three different $\tau$ yield similar performance gap between CENTROBIND and FABIND , implying that the advantage of

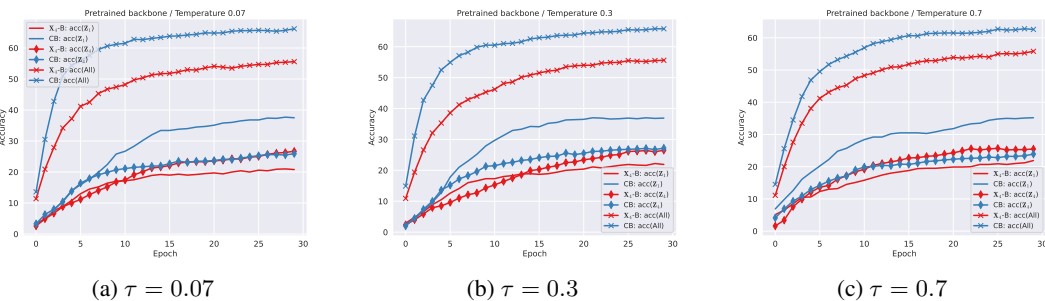

(a) $\tau = 0.07$    (b) $\tau = 0.3$    (c) $\tau = 0.7$

Figure 7: Classification accuracy on GMM synthetic dataset. The embeddings are learned with different temperature parameter $\tau \in \{0.07, 0.3, 0.7\}$.

CENTROBIND is robust to hyperparameter. This further strengthens our claim that CENTROBIND overcomes the inherent limitation of FABIND.

**Complexity.** CentroBind introduces only one additional InfoNCE term per batch—the interaction between the learned centroid and the modality embeddings—so computational complexity rises merely from $O(M)$ to $O(M + 1)$ for $M$ modalities. In memory, it stores a single extra vector (the current-batch centroid), adding a negligible constant overhead. Consequently, training and inference costs remain practically unchanged from FABind.

Table 5: Classification accuracies presented in Figure 10. In the experiment in Figure 10a and 10b, $\mathbf{X}_1, \mathbf{X}_2,$ and $\mathbf{X}_3$ are very noisy, and $\mathbf{X}_4$ is highly informative. In the experiment in Figure 10c and 10d, $\mathbf{X}_1$ and $\mathbf{X}_2$ are very noisy, and $\mathbf{X}_3$ and $\mathbf{X}_4$ are highly informative. We choose $\mathbf{X}_4$ for FABIND for the best-fixed anchor modality for both cases. The weighted average method uses prior knowledge of modality quality to determine the weights for each modality. Random anchor method without intra-learning uses a randomly chosen modality as an anchor for each iteration under a fixed anchor encoder, while with intra-learning we train intra-modal learning by not freezing the anchor encoder.

| Backbone | Method | Figure 10a and 10b | | | Figure 10c and 10d | | |
|---|---|---|---|---|---|---|---|
| | | $\mathbf{X}_2$ | $\mathbf{X}_4$ | $\mathbf{X}_1, \cdots, \mathbf{X}_4$ | $\mathbf{X}_2$ | $\mathbf{X}_4$ | $\mathbf{X}_1, \cdots, \mathbf{X}_4$ |
| | $\times$ | 0.115 | 0.296 | 0.566 | 0.099 | 0.256 | 0.537 |
| | FABIND-$\mathbf{X}_4$ | 0.124 | 0.297 | **0.639** | 0.115 | 0.263 | 0.540 |
| | CENTROBIND | 0.131 | **0.363** | 0.618 | **0.116** | 0.336 | 0.563 |
| Pre-trained | Weighted average | 0.133 | 0.342 | 0.609 | 0.102 | 0.338 | 0.574 |
| | Random + intra learning | 0.131 | 0.347 | 0.613 | 0.105 | 0.353 | 0.554 |
| | Random anchor | **0.147** | 0.359 | 0.619 | 0.097 | 0.327 | 0.579 |
| | Median (coordinate-wise) | 0.134 | **0.363** | 0.634 | 0.112 | **0.375** | **0.582** |
| | $\times$ | 0.092 | 0.114 | 0.487 | 0.067 | 0.176 | 0.465 |
| | FABIND-$\mathbf{X}_4$ | 0.131 | 0.143 | 0.575 | 0.112 | 0.153 | 0.523 |
| | CENTROBIND | 0.132 | **0.355** | **0.626** | **0.113** | **0.336** | 0.559 |
| Random | Weighted average | 0.115 | 0.347 | 0.619 | 0.109 | **0.336** | 0.556 |
| | Random + intra learning | 0.132 | 0.333 | 0.602 | 0.104 | 0.309 | 0.562 |
| | Random anchor | **0.145** | 0.354 | 0.618 | 0.097 | 0.324 | 0.552 |
| | Median (coordinate-wise) | 0.137 | 0.347 | 0.612 | 0.112 | 0.330 | **0.565** |

**Comparison with other adaptive anchor generation.** We compare the centroid-based adaptive anchor method with other potential approaches, such as weighted average, random anchor fixing, and component-wise median. Figure 10 illustrates the accuracies of each method under scenarios where modalities are unevenly distributed. Specifically, we create 4 modalities with differing quality levels. In experiments (a) and (b) of Figure 10, $\mathbf{X}_1, \mathbf{X}_2,$ and $\mathbf{X}_3$ are set as highly uninformative, while $\mathbf{X}_4$ represents a high-quality dataset. Conversely, experiments (c) and (d) use $\mathbf{X}_1$ and $\mathbf{X}_2$ as poor-quality datasets, while $\mathbf{X}_3$ and $\mathbf{X}_4$ are high-quality datasets.

Table 6: Classification accuracy evaluated on each modality (training and evaluation modalities are the same) with MUStARD dataset. Asterisk* denotes different backbone encoders and pretraining settings.

| Method | Modality | Sar-1 | Spk-1 | Spk-3 | Spk-5 |
|---|---|---|---|---|---|
| FABIND | | 0.606 | 0.219 | 0.458 | 0.632 |
| UniBind | | 0.600 | 0.214 | 0.412 | 0.569 |
| AudioCLIP* | $\mathcal{T}$ | 0.488 | 0.155 | 0.280 | 0.388 |
| ViT-Lens* | | 0.543 | 0.172 | 0.342 | 0.472 |
| CENTROBIND | | **0.667** | **0.287** | **0.507** | **0.642** |
| FABIND | | 0.668 | 0.375 | 0.587 | 0.691 |
| UniBind | | 0.658 | 0.381 | 0.641 | 0.770 |
| AudioCLIP* | $\mathcal{V}$ | 0.504 | 0.110 | 0.275 | 0.414 |
| ViT-Lens* | | **0.697** | **0.586** | **0.738** | **0.797** |
| CENTROBIND | | 0.670 | 0.380 | 0.609 | 0.726 |
| FABIND | | 0.639 | 0.201 | 0.457 | 0.599 |
| UniBind | | 0.633 | 0.272 | 0.528 | 0.691 |
| AudioCLIP* | $\mathcal{A}$ | 0.525 | 0.158 | 0.343 | 0.454 |
| ViT-Lens* | | **0.686** | **0.396** | **0.664** | **0.8** |
| CENTROBIND | | 0.616 | 0.234 | 0.461 | 0.609 |
| FactorCL* | | 0.699 | - | - | - |
| SimMMDG* | | 0.725 | - | - | - |
| FABIND | $\mathcal{V},\mathcal{A},\mathcal{T}$ | 0.678 | 0.343 | 0.554 | 0.677 |
| UniBind | $(\mathcal{V},\mathcal{A},\mathcal{T})$ | 0.646 | 0.383 | 0.622 | 0.764 |
| AudioCLIP* | | 0.530 | 0.119 | 0.261 | 0.378 |
| ViT-Lens* | | **0.731** | **0.506** | **0.736** | **0.812** |
| CENTROBIND | | 0.704 | 0.346 | 0.594 | 0.733 |

For the weighted average method (denoted as WAB in Figure 10), we assign weights based on modality quality: $(0.2, 0.2, 0.2, 1)$ for experiments (a) and (b), and $(0.2, 0.2, 0.8, 0.8)$ for experiments (c) and (d). These weights correspond to the information rate of each modality.

For the random modality dynamic anchor method (denoted as RB in Figure 10), we randomly select one modality as the dynamic anchor at each iteration, with the anchor encoder frozen. To investigate the impact of intra-modal learning, we also conduct experiments with a random anchor that includes intra information learning (denoted as RB+Intra). In this case, the anchor modality is randomly selected at each iteration, and the anchor encoder is not frozen, allowing all encoders to be trained.

Since the median is more robust to outliers than the average (Lopuhaä & Rousseeuw, 1991), we additionally evaluate the case of a median-based dynamic anchor. In high-dimensional spaces, rather than in the univariate case, a coordinate-wise median can be used as a naive generalization of the univariate median to the multivariate setting, preserving its robustness to outliers. We assess the dynamic anchor binding method using the coordinate-wise median approach (denoted as MB in Figure 10). Specifically, for the median anchor, we compute the $j$th coordinate of the $i$th anchor as $\boldsymbol{a}_{i,j} = \mathrm{Median}(\boldsymbol{z}_{1,i,j}, \boldsymbol{z}_{2,i,j}, \cdots, \boldsymbol{z}_{M,i,j})$, where $\boldsymbol{z}_{m,i,j}$ denotes the $j$th coordinate of the embedding for the $i$th sample in modality $m$. For improved readability, we summarize the final accuracies for each method and modality in Table 5.

This scenario, where modal distributions are uneven, is commonly referred to as the *modality imbalance* problem (Du et al., 2023; Peng et al., 2022; Zhang et al., 2024). Intuitively, in the presence of modality imbalance, the centroid may produce suboptimal dynamic anchor constructions, and other methods, such as weighted averages, might yield better results. Nevertheless, CENTROBIND consistently performs better or comparably to weighted average methods, demonstrating its robustness to the modality imbalance problem.

From these experiments, we conjecture that the specific dynamic anchor generation method may not significantly impact final performance, provided that all encoders are well-trained during the process.

Addressing the modality imbalance problem typically requires additional information, such as domain knowledge, labels, or downstream task insights. Since this work focuses on multi-modal alignment under contrastive learning, we do not assume such information is available. We therefore leave the exploration of the modality imbalance problem for dynamic anchor generation as a direction for future work.

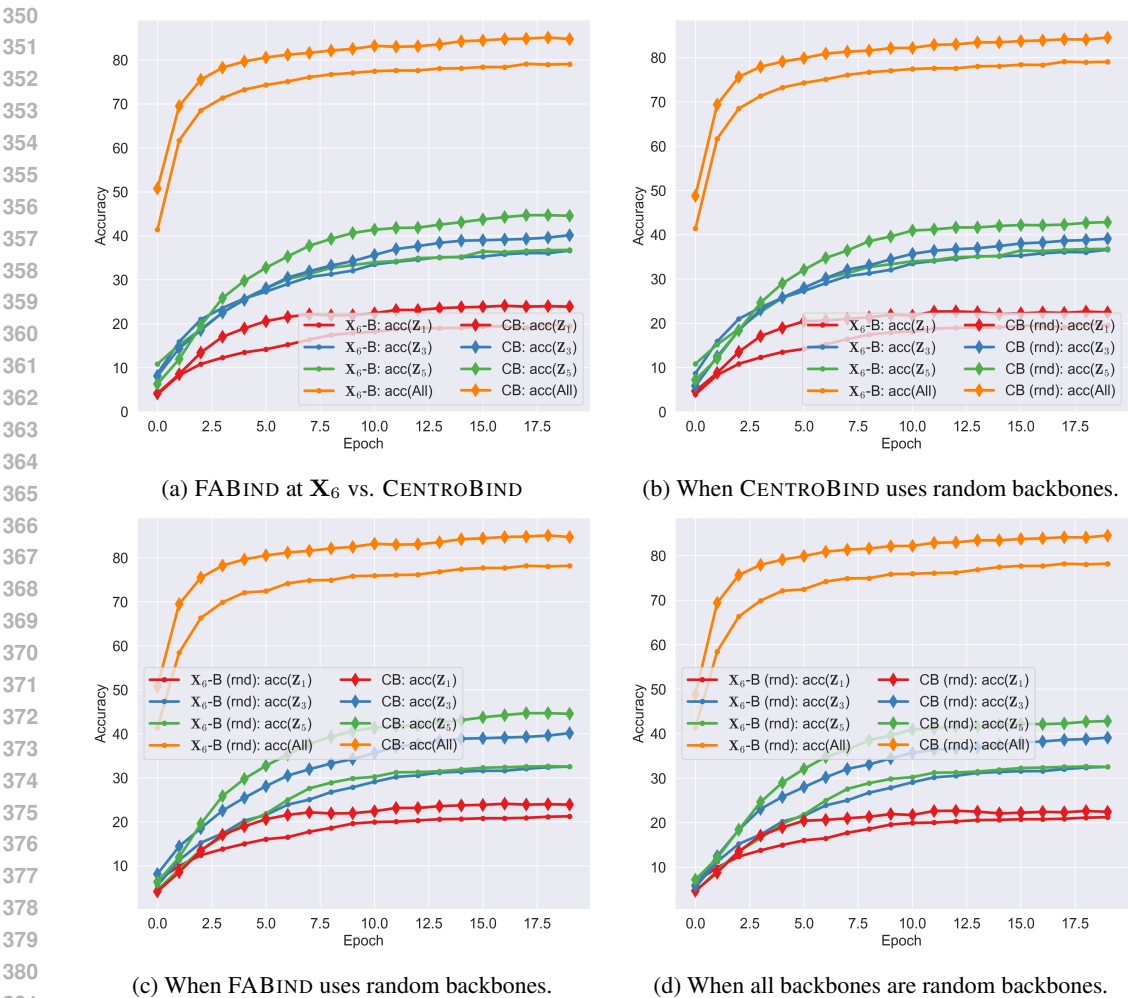

(a) FABIND at $\mathbf{X}_6$ vs. CENTROBIND

(b) When CENTROBIND uses random backbones.

(c) When FABIND uses random backbones.

(d) When all backbones are random backbones.

Figure 8: Experiment results with synthetic dataset of $M = 6$ modalities. Abbreviation: $\mathbf{X}_i$-B or CB: applying FABIND method to backbones with anchor $\mathbf{X}_i$ or applying CENTROBIND; $acc(\mathbf{Z}_i)$ or $acc(All)$: accuracy of $\mathbf{Z}_i$ or of concatenated embeddings $(\mathbf{Z}_1, \cdots, \mathbf{Z}_M)$; (rnd): if random backbones are used for $\mathbf{X}_i$-B or CB.

## C.2 EXPERIMENTS ON MUSTARD

**Training details.** We utilize Low-Rank Adaptation (Hu et al., 2022) for training CENTROBIND and FABIND, enhancing training efficiency and achieving impressive results with fewer iterations. For the backbones in FABIND and CENTROBIND, we use the pretrained VideoMAE (Tong et al., 2022) for video data, the pretrained WaveLM (Chen et al., 2022) for audio data, and the pretrained BERT (Devlin et al., 2019) for text data. For parameter settings, we set a learning rate of $0.001$, the AdamW optimizer (Loshchilov & Hutter, 2019) with a batch size of $16$, and a temperature of $0.3$ for InfoNCE. Training CENTROBIND requires augmentation. We augment video frames with various transformations, including random perspective shifts, random flips and rotation, color jitter, Gaussian blur, and auto-contrast adjustment. For the audio modality, we apply a low-pass filter, speed changes, echo effect, room impulse response convolution, and background noise. For the text modality, we generate paraphrased sentences using the Phi-3 language model served using Ollama [4].

**UniBind** We evaluate UniBind as a baseline method, using LLM-generated descriptions as the anchor modality. Specifically, UniBind generates descriptions for each modality using a large

---

[4] https://ollama.com/library/phi3

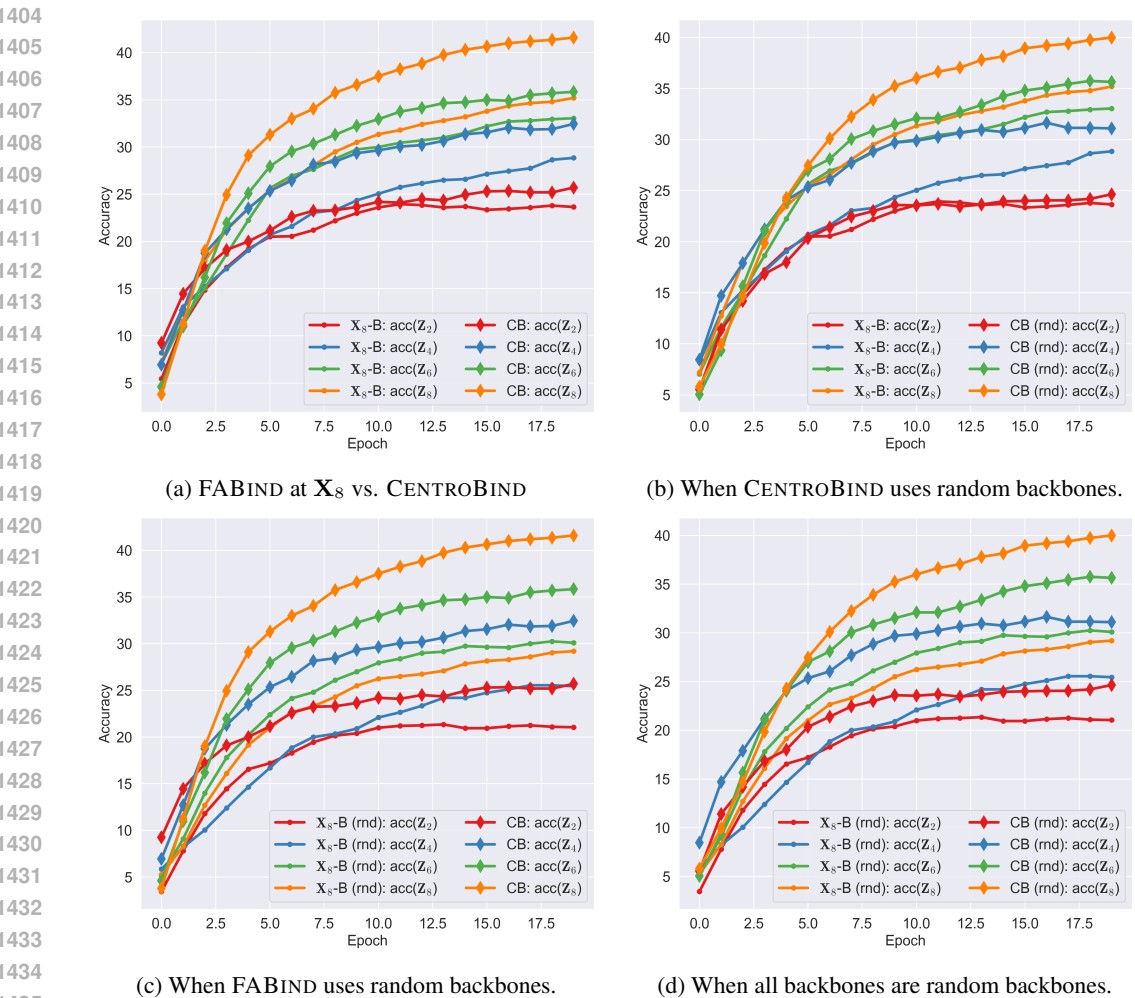

(a) FABIND at $\mathbf{X}_8$ vs. CENTROBIND

(b) When CENTROBIND uses random backbones.

(c) When FABIND uses random backbones.

(d) When all backbones are random backbones.

Figure 9: Experiment results with synthetic dataset of $M = 8$ modalities. Abbreviation: $\mathbf{X}_i$-B or CB: applying FABIND method to backbones with anchor $\mathbf{X}_i$ or applying CENTROBIND; acc($\mathbf{Z}_i$) or acc(All): accuracy of $\mathbf{Z}_i$ or of concatenated embeddings $(\mathbf{Z}_1, \cdots, \mathbf{Z}_M)$; (rnd): if random backbones are used for $\mathbf{X}_i$-B or CB.

language model (LLM), ensuring that every modality is paired with corresponding descriptions. These descriptions collectively form a knowledge base, and UniBind optimizes the InfoNCE loss between each modality and its paired description from the knowledge base. In this framework, the anchor modality is the LLM-augmented representation. It is important to note that the LLM-generated descriptions for different modality pairs can vary, which may hinder effective multi-modal alignment (see Table 2). In our experiments, we generate descriptions for video and audio modalities using the VideoLLaMA2.1-7B-AV audio-visual model from VideoLLaMA2 (Cheng et al., 2024), and for the text modality, we use the Qwen2.5-32B-Instruct model from Qwen2.5 (Team, 2024). We evaluate UniBind's performance in two settings: standard classification accuracy (Table 6) and zero-shot cross-modal classification (Table 2).

**AudioCLIP** We employ AudioCLIP (Guzhov et al., 2022), which aligns image, text, and audio representations into a unified multi-modal space. To extend its capabilities to the video modality in our experiments, we adapt AudioCLIP to extract embeddings for video, audio, and text modalities using a pretrained model. For audio, we follow AudioCLIP's approach, padding audio samples to ensure uniform input sizes. For text, we utilize its pretrained settings, truncating tokenized text to 77 tokens, which only occurs in one instance. For the video modality, we use the center frame as

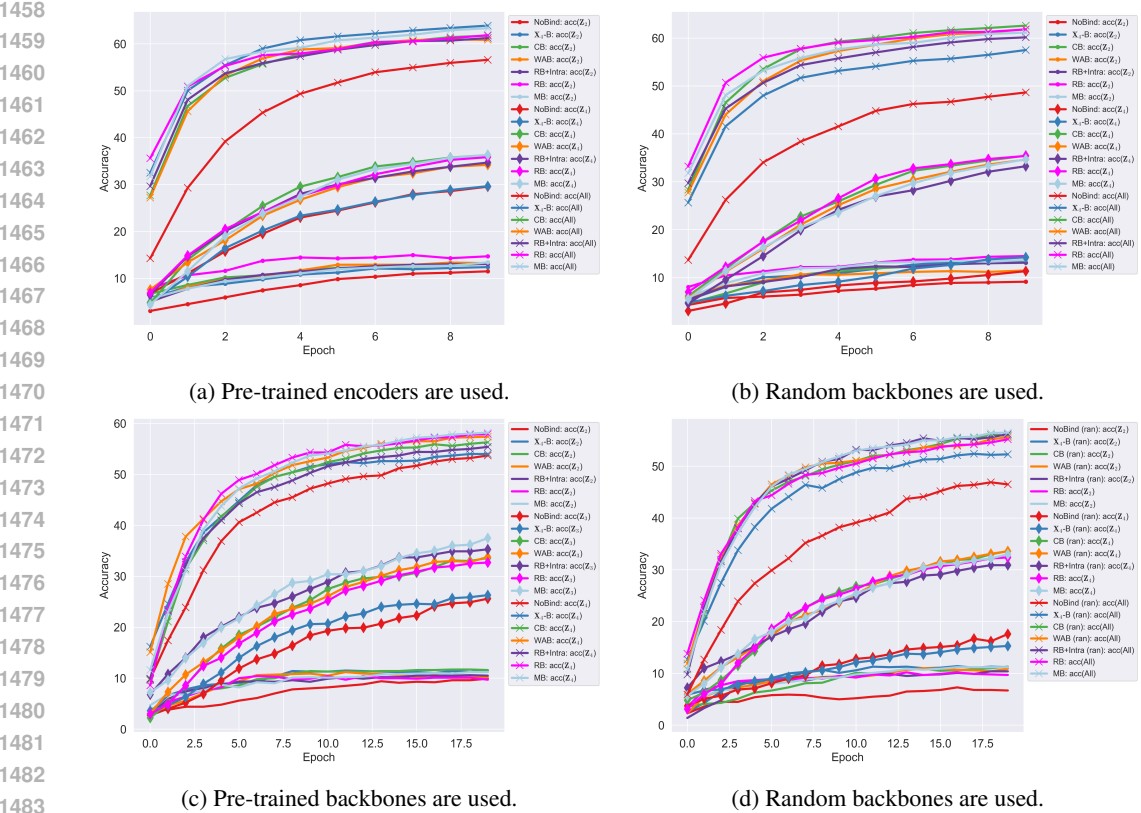

Figure 10: Comparison of other dynamic anchor generation methods. (a) and (b): Modal qualities are set to $(0.2, 0.2, 0.2, 1)$. (c) and (d): Modal qualities are set to $(0.2, 0.2, 0.8, 0.8)$. Abbreviation: $\mathbf{X}_i$-B or CB: applying FABIND method to backbones with anchor $\mathbf{X}_i$ or applying CENTROBIND; WAB: weighted average for dynamic anchor with weight identical to the predefined quality for each modality; RB+Intra: randomly choosing a modality for a dynamic anchor in every iteration and intra information learning; RB: randomly choosing a modality for a dynamic anchor in every iteration; MB: coordinate-wise median for dynamic anchors; $\mathrm{acc}(\mathbf{Z}_i)$ or acc(All): accuracy of $\mathbf{Z}_i$ or of concatenated embeddings $(\mathbf{Z}_1, \cdots, \mathbf{Z}_M)$; (ran): if random backbones are used.

a representative image sample. Finally, embeddings from all three modalities are concatenated for downstream tasks.

**ViT-Lens** In our experiments, we leverage the pretrained models from ViT-Lens to extract embeddings for audio, text, and video modalities. We generally follow the example code provided by the authors.[5] Note that we select the center frame image from the video to extract the embedding.

**Classification results.** In contrast to the cross-modal retrieval results in Table 1 and zero-shot cross-modal classification in Table 2, Table 6 presents the classification accuracy of FABIND, UniBind, and CENTROBIND for each modality as well as for multi-modal scenarios. Specifically, embeddings are extracted using the binding methods, and a simple decoder is trained to classify the embeddings. In Table 6, we report the sarcasm and speaker classification accuracies of decoders trained and evaluated on the same modality.

For sarcasm detection, CENTROBIND generally outperforms other baseline methods. While UniBind performs poorly in cross-modal classification, it achieves better performance in speaker classification compared to others. This improvement is due to the LLM-augmented descriptions, which provide additional knowledge (from LLMs) to the embeddings. Notably, UniBind utilizes 4 modalities, whereas FABIND and CENTROBIND only use 3, which could penalize the performance of FABIND

---

[5]https://github.com/TencentARC/ViT-Lens

and CENTROBIND . Nevertheless, CENTROBIND consistently outperforms FABIND. Moreover, our method can also incorporate LLM-augmented descriptions as an additional modality, potentially improving its performance further.

Although a direct comparison is not feasible, we also include the sarcasm detection accuracy of FactorCL (Liang et al., 2024b), SimMMDG (Dong et al., 2023), AudioCLIP (Guzhov et al., 2022), and ViT-Lens (Lei et al., 2024) for reference. ViT-Lens, in particular, achieves higher performance than CENTROBIND due to its use of larger backbone encoders, such as Vision Transformer (ViT) (Khan et al., 2022) and pretraining on extremely large-scale datasets. However, since ViT-Lens can be considered a variant of FABind, applying our dynamic anchor method could further improve its performance. Specifically, ViT-Lens uses a pretrained CLIP model as the anchor encoder, while the other non-anchored modalities use pretrained ViT models with modality adaptation layers. Within our framework, CENTROBIND could adopt the pretrained Vision Transformer as backbone encoders, potentially enhancing its performance further.

### C.3 EXPERIMENTS WITH PRE-TRAINED BACKBONE

So far, the superiority of CENTROBIND is verified for controllable synthetic and MUStARD datasets. Although such experiments and theoretical analysis show the effectiveness of CENTROBIND, we further evaluate and compare the performance of CENTROBIND and FABIND to validate in additional scenarios. In particular, we consider the following cases:

1. Bi-modal scenario → We compare performance in bi-modal datasets, DreamBooth (Ruiz et al., 2023), AVE (Tian et al., 2018), AudioSet (Gemmeke et al., 2017), UR-FUNNY (Hasan et al., 2019).

2. When a large-scale powerful backbone is available → We compare performance given the pre-trained ImageBind backbone.

3. When strong anchor modality is available → We compare performance with the image modality as anchor.

We conduct fine-tuning the ImageBind and CENTROBIND on DreamBooth (Ruiz et al., 2023), AVE (Tian et al., 2018), AudioSet (Gemmeke et al., 2017), UR-FUNNY (Hasan et al., 2019) datasets. For datasets containing video modality instead of image, we extract the middle frame from the video and use the middle frame as image sample. Audio is also extracted from video if audio modality does not exist in original dataset. The evaluation is conducted through a retrieval task, effectively measuring the quality of the learned unified embedding space.

After fine-tuning, we measure the top-1 and top-5 retrieval accuracy on the test dataset. The optimization process uses the AdamW optimizer with the following hyperparameters: a batch size of 16, a learning rate of $5.0 \times 10^{-6}$, a weight decay of $10^{-4}$, and a temperature parameter set to 0.07. We fine-tune the models until their validation accuracies converge to certain value. In this experiment, we omit the text augmentation. Instead, we create a centroid-based dynamic anchor using the embeddings of augmented images and original text.

More specifically, we begin by loading the pre-trained ImageBind model (Girdhar et al., 2023) and fine-tune its encoders. For fine-tuning, we employ Low-Rank Adaptation (LoRA) (Hu et al., 2022) with a rank of 4, resulting in 5.1-5.4 million trainable parameters (depending on chosen modalities) out of a total of 1.2 billion parameters. For the experiment on AudioSet, we fine-tune the models on the Balanced train subset of AudioSet, provided in official site. For datasets provided without train-test split, we randomly split the dataset with $0.8 : 0.2$ ratio before finetuning. The results in Figure 11 and Table 3 show that CENTROBIND outperforms ImageBind even with a strong pretrained backbone and the image modality. Moreover, CENTROBIND achieves additional gains in bimodal settings, suggesting it mitigates FABIND 's inherent limitations across most scenarios. We anticipate that the performance gap would widen if the backbone were weaker, if the image modality were excluded, or if additional modalities were introduced. These findings corroborate our analysis of dynamic anchor binding, highlighting its effectiveness in enhancing multimodal representations.

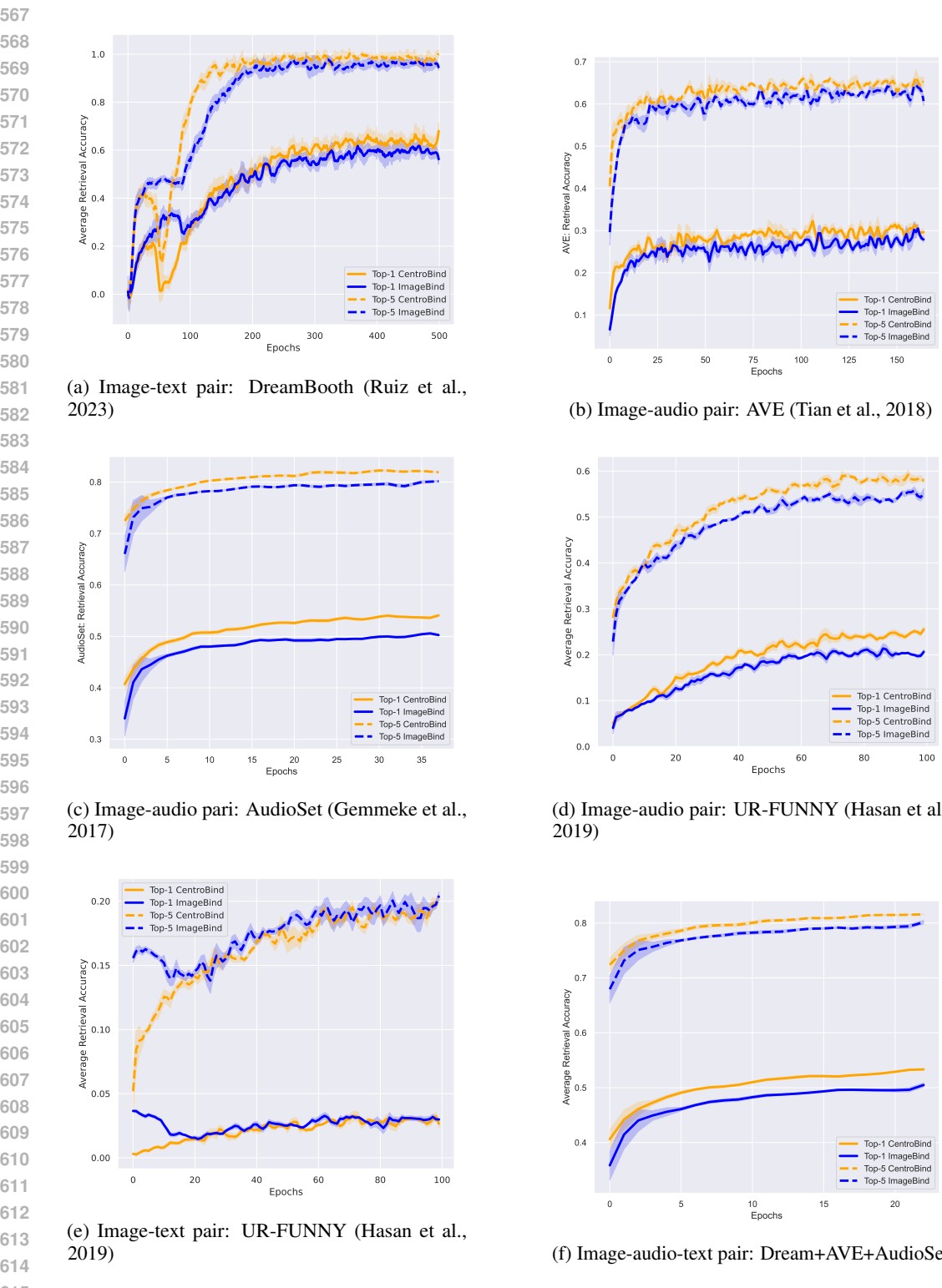

(a) Image-text pair: DreamBooth (Ruiz et al., 2023)

(b) Image-audio pair: AVE (Tian et al., 2018)

(c) Image-audio pari: AudioSet (Gemmeke et al., 2017)

(d) Image-audio pair: UR-FUNNY (Hasan et al., 2019)

(e) Image-text pair: UR-FUNNY (Hasan et al., 2019)

(f) Image-audio-text pair: Dream+AVE+AudioSet

Figure 11: We evaluate cross-modal retrieval using the pretrained ImageBind backbone, fine-tuned on each dataset with both ImageBind and CentroBind. Apart from the image–text pair in the UR-FUNNY dataset—where performance is identical—CentroBind surpasses ImageBind in every other setting.

