# OpenReview forum: "Anchors Aweigh! Sail for Optimal Unified Multi-Modal Representations"
_ICLR.cc/2026/Conference — ICLR 2026 Conference Withdrawn Submission_

### Official Review · Reviewer_Ux9F · 2025-10-31

**Soundness:** 2
**Presentation:** 2
**Contribution:** 2
**Rating:** 2
**Confidence:** 4

**Summary:**

This paper revisits the challenge of constructing a unified representation space across multiple modalities (e.g., text, image, audio) for multimodal learning. Existing methods, such as ImageBind, rely on a fixed anchor modality (e.g., images or text) as the target space into which all other modalities are aligned, thus leading to reliance on the anchor modality and loss of intra-model interaction. To overcome these issues, the paper proposes CENTROBIND by constructing a centroid-based, dynamically updated anchor embedding to act as the anchor for all modalities. Empirical results on synthetic and real-world datasets demonstrate consistent improvements over FABIND baselines.

**Strengths:**

- This paper studies a practical problem where the fixed anchor could be limited in some cases.
- A theoretical framework is proposed to support the method.

**Weaknesses:**

- The anchor generation strategy in CentroBind, which averages modality centroids, may not be robust when different modalities exhibit varying information densities. Modalities containing more or less discriminative information could disproportionately influence the centroid, potentially leading to biased or unbalanced representations.
- The proposed method relies on high independence of modalities, which is not true in the real world. When modalities are highly correlated or exhibit strong synergy, the centroid could be overly biased toward certain modalities. Such correlation might hinder optimization, preventing the model from capturing true multimodal relationships. The paper does not clearly address how the optimization process mitigates these dependencies.
- The combined use of multiple InfoNCE losses may introduce convergence instability or slow training. The effects of varying temperature parameters or weighting schemes among loss terms are not discussed, limiting understanding of the loss optimization dynamics and robustness.
- The theoretical analysis appears to assume ideal conditions, leaving unclear the boundary cases where CentroBind may underperform—such as under uneven data distributions, severe modality imbalance, or non-independent modality structures. The generalizability of the theoretical claims thus remains uncertain.

**Questions:**

Please see the weaknesses part.

---

### Official Review · Reviewer_6D21 · 2025-11-01

**Soundness:** 3
**Presentation:** 3
**Contribution:** 3
**Rating:** 4
**Confidence:** 4

**Summary:**

The paper is motivated by a problem in “bind-everything-to-images” style models (ImageBind, LanguageBind, etc.): if you always align other modalities to one fixed anchor (usually vision), the final joint space can only be as rich as that anchor, so any information that lives mostly in audio, text, or another modality is suppressed. Even worse, correlations between two non-anchor modalities (say, audio↔text) are never explicitly optimized, because the loss only looks at anchor↔others. So the authors ask: can we build a unified space that is co-defined by all modalities, not dictated by just one?

Their method, CENTROBIND, replaces the fixed anchor with an adaptive, batch-wise anchor: for each training batch, they compute a centroid from the available modalities embeddings and use that as the “anchor” everyone aligns to. Then they apply a contrastive / InfoNCE-style objective that pulls every modality toward this centroid, so the shared space is shaped by what actually appears in the data in that batch, not by a single handpicked modality. Because the center is recomputed each time, it can drift to accommodate different modality combinations and preserve modality-specific signals.

Empirically, on both synthetic and real multimodal setups, CENTROBIND outperforms fixed-anchor baselines, matching the theory that multi-modal alignment should be symmetric (all-to-one adaptive center) rather than asymmetric (all-to-one fixed image space).

**Strengths:**

- Theoretical justification for the problem is solid, in standard multimodal contrastive learning the choice of anchor modality imposes a fixed ceiling

- The propose CENTROBIND method is simple: compute a per-batch centroid and align to it, no additional architecture, so in principle you can drop it into existing multimodal contrastive setups

**Weaknesses:**

- Assumption that a single centroid per batch is a good proxy for the "true" shared semantics.
- The method is also batch-dependent: the quality and stability of the anchor will depend on what modalities are present and how balanced the batch is.
- Empirical evaluation is very limited, only consisting of results on a synthetic dataset and some limited set of tasks like sarcasm and speaker classification, dreambooth (image editing ?). Audioset is the only result comparable to prior baseline papers.

- Some relevant references to related work are missing and should be discussed in paper:
1. Humam Alwassel, Dhruv Mahajan, Bruno Korbar, Lorenzo Torresani, Bernard Ghanem, and Du Tran. Self-supervised learning by cross-modal audio-video clustering. NeurIPS 2020.
2. Brian Chen, Andrew Rouditchenko, Kevin Duarte, Hilde Kuehne, Samuel Thomas, Angie Boggust, Rameswar Panda, Brian Kingsbury, Rogerio Feris, David Harwath, et al. Multimodal clustering networks for self-supervised learning from unlabeled videos. ICCV 2021.
3. Sirnam Swetha, Mamshad Nayeem Rizve, Nina Shvetsova, Hilde Kuehne, and Mubarak Shah. Preserving modality structure improves multi-modal learning. ICCV 2023.

**Questions:**

Please provide results for all the datasets used in the ImageBind paper which is the main baseline. I am willing to raise my rating if comprehensive evaluation is demonstrated. Also add more recent results for models like LanguageBind and InternVideo2.

---

### Official Review · Reviewer_EEA3 · 2025-11-01

**Soundness:** 2
**Presentation:** 2
**Contribution:** 2
**Rating:** 4
**Confidence:** 4

**Summary:**

The paper presents CentroBind, a framework that adaptively determines the anchor modality for performing multimodal alignment. The paper provides a theoretical analysis showing that this approach can capture intra-modal, inter-modal, and multi-modal alignment components, and the empirical experiments demonstrate that CentroBind outperforms FABind baselines and other recent multi-modal alignment methods on retrieval and zero‐shot classification tasks.

**Strengths:**

- Clear and Well-Structured: The paper is well-organized, with detailed explanations of the preliminary, intuition, and methodology.

- Superiority in Alignment: The experimental results demonstrate that the proposed method achieves the best performance on the cross-modal retrieval and classification tasks compared to the baselines.

**Weaknesses:**

- The paper provides clear intuition but presents the preliminary and methodological sections in an overly complex manner. I suggest that the authors reorganize the presentation flow to enhance readability and logical coherence. From my perspective, it is not necessary to include too many theoretical derivations or formal statements in the main text—these could be moved to the appendix, while keeping the main body focused on the core ideas, motivation, and empirical insights.

- If some modalities’ encoders are significantly stronger or weaker, then the centroid might be dominated by high-quality modality embeddings, thus implicitly reintroducing an anchor bias. While the authors mention weighted aggregation as a workaround, empirical quantification of this phenomenon is minimal.

- The technique introduced in this paper is relatively trivial and does not address the essential issue of anchor bias. While the proposed modification may bring marginal improvements, it fails to fundamentally eliminate the dependence on specific modalities or to provide a principled mechanism for balancing heterogeneous modality contributions.

- The paper does not include experimental comparisons with other recent multimodal alignment methods, such as TRIANGLE [1] and GRAM [2]. Including these baselines would provide a stronger empirical validation of the proposed method's effectiveness.


[1] A TRIANGLE Enables Multimodal Alignment Beyond Cosine Similarity, NeurIPS 2025

[2] Gramian multimodal representation learning and alignment, ICLR 2025

**Questions:**

See Weaknesses

---

### Official Review · Reviewer_Yya1 · 2025-11-02

**Soundness:** 2
**Presentation:** 2
**Contribution:** 2
**Rating:** 2
**Confidence:** 3

**Summary:**

This paper proposes CentroBind, an adaptive anchor binding method for multi-modal representation learning.
The authors argue that fixed-anchor binding (FABind) methods suffer from over-reliance on a single anchor modality, loss of intra-modal information, and failure to capture shared information among non-anchor modalities.
CentroBind replaces the fixed anchor with a centroid-based anchor computed from all available modalities.
The method is evaluated on synthetic and real-world datasets and is shown to outperform FABind and several other baselines in tasks such as cross-modal retrieval and classification.

**Strengths:**

1. The paper is clearly written and motivated by a relevant problem in multi-modal learning — the bias and inefficiency of fixed-anchor alignment.
2. The proposed centroid-based adaptive anchor idea is simple, easy to implement, and potentially applicable to other multi-modal frameworks.
3. Experiments are conducted on both synthetic and real-world datasets, covering multiple modalities and tasks.

**Weaknesses:**

1. The theoretical novelty is relatively weak.

(1) The key idea—constructing a centroid anchor from multiple modalities—is a minor variation of existing multi-modal alignment formulations.

(2) Similar concepts of adaptive or learned anchors have been discussed in OmniBind[1] and UNIALIGN[2].

(3) The mathematical derivations in Section 3 mostly restate standard InfoNCE lower-bound properties without offering new theoretical insights or proofs that go beyond prior work. The formal results (Theorem 3.1, Propositions 2.2–2.3) repackage well-known properties of mutual information and data-processing inequality.

2. While the authors evaluate on several datasets, the experimental validation is not convincing for a top-tier conference.

(1) Baselines are not sufficient. The comparisons include FABIND, UniBind, AudioCLIP, and ViT-Lens, but exclude more recent and stronger baselines such as OmniBind [1] and UNIALIGN [2], which are both highly relevant and directly comparable.

(2) Lack of large-scale evaluation. All reported results are on small or medium datasets (MUStARD, AudioSet subsets). For a method claiming general multi-modal unification, evaluations on larger benchmarks are expected.

3. CENTROBIND introduces extra computation for per-batch centroid construction and additional InfoNCE terms.
However, the paper does not provide complexity analysis or runtime comparison with strong baselines such as ViT-Lens and UniBind.
Without this, it is unclear whether the method scales to high-dimensional multi-modal tasks.

4. The manuscript contains several minor formatting and reference errors (e.g., “Figure ??” on page 6, line 273).

5. Appendix references are frequently cited for essential content (algorithms, proofs, and ablations), making it difficult to assess the main claims within the body of the paper.

[1] Lyu, Yuanhuiyi, Xu Zheng, Dahun Kim, and Lin Wang. "Omnibind: Teach to build unequal-scale modality interaction for omni-bind of all." arXiv preprint arXiv:2405.16108 (2024).

[2] Zhou, Bo, Liulei Li, Yujia Wang, Huafeng Liu, Yazhou Yao, and Wenguan Wang. "UNIALIGN: Scaling Multimodal Alignment within One Unified Model." In Proceedings of the Computer Vision and Pattern Recognition Conference, pp. 29644-29655. 2025.

**Questions:**

1. Could the authors include direct experimental comparisons with OmniBind [1] and UNIALIGN [2]?
2. Have the authors considered learning the centroid weights (e.g., modality-dependent coefficients) instead of computing a simple mean?
3. What happens if modalities are partially missing during training or inference — can the adaptive anchor still be constructed robustly?

---

### Comment · Area_Chair_g9eV · 2025-11-26

Dear authors,
we note that no author response has been posted during the author response window (Discussion Period: Nov 12 – Dec 3, 2025). To ensure a fair review, please post a reply addressing the reviewers' main concerns on this forum by Dec 3, 2025.

---

### Note · Authors · 2025-12-03

I have read and agree with the venue's withdrawal policy on behalf of myself and my co-authors.